# Benchmarking Stochastic Interpolants for Physical Emulation

## Abstract

Generative models have recently emerged as powerful surrogates for physical systems, demonstrating increased accuracy, stability, and/or statistical fidelity. However, most approaches rely on iteratively denoising a Gaussian, a choice that may not be the most effective for autoregressive prediction tasks in PDEs and dynamical systems such as climate. Given the proximity of current and future distributions in these tasks, we consider generative models based on stochastic interpolants, which can directly learn maps between arbitrary densities. To evaluate this framework, we benchmark a variety of generative models across fluid and climate systems. Our experiments suggest that stochastic interpolants can use fewer sampling steps and produce more accurate predictions than models relying on transporting Gaussian noise. We also observe broad trends across generative models, such as the ability to exchange deterministic accuracy, spectral consistency, and probabilistic calibration through sampling. Given these results, this study establishes stochastic interpolants as a competitive baseline for physical emulation and gives insight into the abilities of different generative modeling frameworks.

## 1 Introduction

Generative models have recently become a promising class of models for physical systems. Empirically, diffusion models have demonstrated better accuracy and stability (Lippe et al., 2023), capable of resolving finer details than deterministic baselines (Oommen et al., 2025). In addition, diffusion models can be more effective at capturing the underlying statistics of physical systems, such as in turbulence (Lienen et al., 2023; Molinaro et al., 2025) or in weather forecasting and climate prediction (Price et al., 2024; Cachay et al., 2023). Overall, these capabilities are supported by studies that benchmark diffusion models in PDE systems (Kohl et al., 2024; Rozet et al., 2025), as well as by continued research that improves their accuracy or inference speed (Bastek et al., 2025; Shehata et al., 2025). Moreover, these models continue to benefit from larger advances in the generative modeling community, such as flow matching or improved samplers (Liu et al., 2022; Lu et al., 2025).

While promising, a key feature of these prior works in PDE or climate modeling is the assumption of a Gaussian prior/source distribution. This is a logical choice for unconditional generation, where we have no prior knowledge about the source distribution and require it to be easily sampled. However, recent work has challenged this assumption for tasks where the source and target distributions are related, such as in image-to-image translation or super-resolution. Ordinarily, these tasks are framed as sampling from a Gaussian and evolving a reverse process conditioned on the source distribution, however, methods such as diffusion bridges or stochastic interpolants seek to directly learn a stochastic process between the source and target distributions (Zhou et al., 2023; Albergo & Vanden-Eijnden, 2023). Directly evolving samples drawn from the source distribution (e.g., blurry images, masked images) to samples from the target distribution (e.g., sharp images, in-painted images) can require fewer sampling steps and produce higher quality samples (Albergo et al., 2024; Zheng et al., 2025). In these cases, it is believed that transporting Gaussian noise to a conditional distribution is both inefficient and more complex than directly mapping the source to the target distribution.

This observation makes stochastic interpolants well-suited as a generative model for physical systems. Specifically, predicting PDE or climate systems is usually framed as an autoregressive task, where future states are predicted based on a current state. While current and future states are often tightly coupled, generative models still predominantly follow the approach of transporting Gaussian

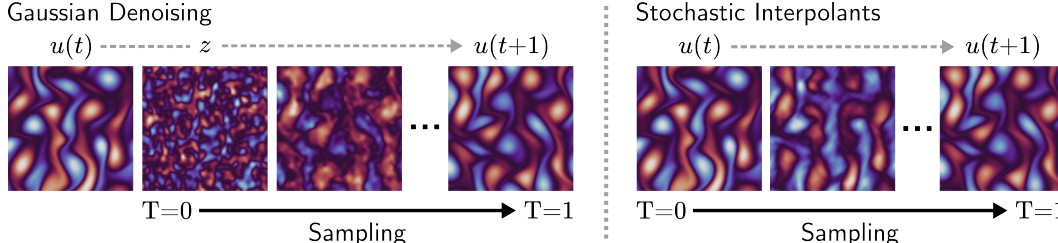

Figure 1: *Left:* Typical generative models rely on transporting Gaussian noise $z$, conditioned on information of the current state $u(t)$ at every timestep $T$. *Right:* Learning a stochastic process to transport current to future states can be more efficient and accurate, where drift estimates are also conditioned on $u(t)$. $T$ denotes time along a stochastic process, while $t$ is the physical time.

noise while conditioning on the current state. To understand if stochastic interpolants or related approaches can help for these tasks, we benchmark a variety of generative models across different physical emulation tasks. We find that interpolant-based models can result in faster or more accurate generative models, but these benefits are dependent on the underlying system, sampling scheme, and task at hand. We provide a conceptual comparison in Figure 1.

## 2 STOCHASTIC INTERPOLANTS

**Definition** We consider a class of generative models that admit arbitrary source and target distributions. These models have various names and instantiations, however, for simplicity, we consider stochastic interpolants since many of these models can be unified under this framework (Albergo et al., 2023; Zhang et al., 2025). Given samples $x_0$ from a source distribution $\rho_0$ and samples $x_1$ from a target distribution $\rho_1$, a stochastic interpolant is defined as a stochastic process $x_t$ (Chen et al., 2024):

$$x_t = I(t, x_0, x_1) + \sigma(t)W_t, \qquad t \in [0,1] \tag{1}$$

The interpolant $I$ satisfies the boundary conditions $I(0, x_0, x_1) = x_0$ and $I(1, x_0, x_1) = x_1$ and the noise coefficient $\sigma(t)$ is defined such that $\sigma(0)W_0 = \sigma(1)W_1 = 0$ and $\sigma(t) > 0$. Furthermore, $W_t$ is a Wiener process, and for sampling we realize $W_t \overset{d}{=} \sqrt{t}z$ by Brownian increments generated from $z \sim N(0, \mathbf{I})$. There are a few useful observations to make. Firstly, the stochastic interpolant produces samples $x_0 \sim \rho_0$ at $t = 0$ and $x_1 \sim \rho_1$ at $t = 1$ by construction. Furthermore, the interpolant $x_t$ maps between the densities $\rho_0, \rho_1$ exactly and in finite time, which is not true for typical DDPMs in PDE and climate domains. Lastly, a stochastic interpolant can be realized by either an ODE or an SDE, which can produce samples $x_t$ at any time $t \in [0,1]$.

**Implementation** Stochastic interpolants have many instantiations, as well as different training and sampling procedures. We consider spatially linear interpolants, where $x_0$ is sampled from a current solution $u(t)$ and $x_1$ is sampled from a future solution $u(t+1)$:

$$x_t = \alpha(t)x_0 + \beta(t)x_1 + \sigma(t)W_t \tag{2}$$

The coefficients $\alpha(t), \beta(t), \sigma(t)$ are chosen to satisfy boundary conditions. Since these coefficients are specified, we can learn the drift $b$ of the stochastic interpolant with a network $b_\theta$ by minimizing the empirical loss on a dataset $\{x^1, x^2, \ldots, x^N\}$. Additionally, we consider conditional generation, where the drift is conditioned in the current state $x_0 = u(t)$:

$$\mathcal{L}_b[b_\theta] = \frac{1}{N}\sum_{i=1}^{N}\left(|b_\theta(t_i, x_{t_i}^i, x_0^i) - (\dot{\alpha}(t)x_0^i + \dot{\beta}(t)x_1^i + \dot{\sigma}(t)W_t)|^2\right) \tag{3}$$

where $t_i \in [0,1]$ is uniformly sampled and $x_{t_i}^i$ is a given data sample $i$ at time $t_i$ along the stochastic interpolant. Intuitively, $b_\theta$ aims to estimate the time derivative of the stochastic interpolant. This is useful during inference, where samples $x_0$ are integrated using conditional drift estimates $b_\theta$ according to the probability flow SDE, starting from $t = 0$ to $t = 1$:

$$dx_t^{SDE} = b_\theta(t, x_t, x_0)dt + \sigma(t)dW_t \tag{4}$$

At $t = 1$, this SDE produces samples from the conditional distribution $p(x_1|x_0)$. Interestingly, this framework also admits a family of SDEs with variable noise coefficients $\hat{\sigma}(t)$ by constructing a modified drift $\hat{b}_\theta(t, x_t, x_0)$ (Chen et al., 2024). Under regularity conditions, SDEs with variable diffusion can be sampled without retraining $b_\theta(t, x_t, x_0)$. In particular, we are interested in a limiting case where the noise coefficient approaches zero, resulting in an ODE (Appendix A). For generative models, this can allow for faster sampling, and we investigate this tradeoff in the context of stochastic interpolants and physical emulation.

In this work, we choose $\alpha(t) = 1 - t$, $\beta_t = t$, and $\sigma(t) = (1 - t)$. Substituting these values into the interpolant $x_t$ verifies that the noise term $\sigma(t)W_t \overset{d}{=} (1 - t)\sqrt{t}z$ vanishes at the boundaries and we recover $x_1$ at $t = 1$ and $x_0$ at $t = 0$. Furthermore, the diffusion term in the SDE vanishes as $t \to 1$. While this overall loss and sampling framework give an exact generative model, in practice, implementation choices can influence how effectively the drift is learned due to numerical and statistical errors, which we further discuss in Appendix E.

## 3 METHODS

### 3.1 DATASETS

**Kolmogorov Flow**  Kolmogorov Flow (KF) is described by the 2D Navier-Stokes equations driven by unidirectional periodic forcing. Although common, this is a fairly challenging task and most PDE surrogates are unstable when rolled out to the training horizon (Lippe et al., 2023; Zhou & Barati Farimani, 2025). Data is generated from APEBench (Koehler et al., 2024) at a resolution of $160 \times 160$ on a domain $(x, y) = [-10, 10]^2$, with the vorticity being recorded. The simulation is saved at a resolution of $\Delta t = 0.2s$ for 100 timesteps, resulting in a rollout from $t = 0$ to $t = 20$ seconds. Initial conditions are sampled from a random truncated Fourier series with 5 modes, and the viscosity $\nu$ is set to $10^{-2}$ to simulate a Reynolds number of approximately $10^2$.

**Rayleigh-Bénard Convection**  Rayleigh-Bénard Convection (RBC) is a phenomenon that describes the mixing of horizontal layers of fluid driven by a temperature gradient. Current PDE surrogates usually struggle since the system is highly chaotic and features a transition between laminar and turbulent regimes. The system is described by its Rayleigh and Prandtl numbers, which govern the convection and diffusivity of the flow. Data is obtained from the Well (Ohana et al., 2025; Burns et al., 2020), which includes 2D simulations on a $512 \times 128$ grid with buoyancy, pressure, and velocity. In addition, we use 100 timesteps with an interval of $\Delta t = 0.5$ seconds and a variety of Rayleigh and Prandtl numbers are used for training and validation.

**PlaSim**  Global 3D atmospheric data are generated from an intermediate-complexity climate model (PlaSim) to evaluate emulators for weather forecasting and climate prediction (Lunkeit et al., 2021; Ragone et al., 2018; Lancelin et al., 2025). PlaSim solves the Navier-Stokes equation on a rotating sphere along with parameterizations for various atmospheric (e.g., moist convection, radiation) and land processes, while the sea surface temperature and sea ice cover are prescribed and vary with a yearly period. Prognostic atmospheric variables (temperature, humidity, zonal and meridional wind) are saved at a resolution of $(128 \times 64 \times 10)$ (latitude, longitude, model level) on an Gaussian horizontal grid. These are then vertically interpolated onto 13 equipressure levels, and the geopotential height is computed using the hydrostatic equation. 8 surface variables, including 2-meter temperature and accumulated precipitation, and 6 forcing variables are also saved. Models are trained on 100 years of data at 6-hour intervals and validated on a year of held-out data, except when evaluating climatological biases, which uses 10 years of data. This results in $\sim$144,000 training samples and $\sim$1,440 validation samples. Additional detail on all datasets can be found in Appendix D.

### 3.2 MODELS

**Overview**  Although the work focuses on generative models, we include a deterministic emulator as a baseline to understand the difficulty of tasks. For PDE tasks, we consider FNO (Li et al., 2021) and for climate tasks, we consider SFNO (Bonev et al., 2023). To benchmark generative models, we consider: denoising diffusion probabilistic models (DDPM) (Ho et al., 2020), denoising diffusion implicit models (DDIM) (Song et al., 2022), elucidated diffusion models (EDM) (Karras

et al., 2022), truncated sampling models (Shehata et al., 2025) (TSM), flow matching (FM) (Lipman et al., 2023), and stochastic interpolants (SI). These frameworks can have many variations, therefore, formulas for the training objective and sampling procedures used are given in Table 14. Furthermore, models are trained to do conditional generation by giving the current state to all models.

Beyond overall frameworks, generative models are also determined by hyperparameters such as the noise schedule. We make an effort to optimize the baseline parameters both by inheriting best practices from prior benchmarks (Kohl et al., 2024; Shehata et al., 2025) and by performing our own studies (Appendix C). Accordingly, we use a linear schedule for DDPM, DDIM, and TSM models. The noise schedule for EDM is reproduced from Karras et al. (2022). For FM models, we use the rectified flow schedule where $x_t = (1-t)x_0 + tz$, which has shown good performance both in image modeling (Esser et al., 2024) and PDE fields (Rozet et al., 2025; Li et al., 2025b). We standardize ODE samplers by using the Euler method and SDE samplers by using the Euler-Maruyama method.

**Autoencoders** To effectively train generative models, we pretrain a latent space using an autoencoder (Rombach et al., 2022). This is effective in reducing computation, however, for PDE problems, this can additionally stabilize rollouts (Rozet et al., 2025; Li et al., 2025a). For PDE tasks, we use a Deep Compression Autoencoder (DCAE) (Chen et al., 2025a), which is based on 2D convolution and residual pooling/unpooling layers. Following Rozet et al. (2025), we apply a saturation function to latent vectors, rather than KL regularization, to avoid arbitrary variance. For climate tasks, we find that KL regularization is beneficial, potentially due to the need for long-term consistency in climate emulators. KF and RBC autoencoders use a compression ratio of $64\times$ and the PlaSim autoencoder uses a compression ratio of $32\times$.

**Architectures** We use a diffusion transformer (DiT) (Peebles & Xie, 2023) as the backbone for all latent-space generative models. Within a given task, the architecture and model size is kept constant across all models. To condition on the diffusion or interpolant timestep, adaptive layer normalization is used. Additionally, the current state of the system is concatenated to the noisy estimate of the future state to enable conditional generation. Moreover, as a result of the compressed latent space, the backbone does not need patchification or sparse attention. Additional details on the autoencoder and diffusion architectures can be found in Appendix E.

**Metrics** We follow standard metrics to evaluate the deterministic and statistical performance of models. For PDE tasks, we use Variance Scaled RMSE (VRMSE), and for weather forecasting, we use latitude-weighted RMSE (lRMSE). A single prediction is used to evaluate pointwise errors, rather than an ensemble, for generative models. Annual climatological biases are also calculated to evaluate the consistency of the climate emulator; these are calculated as the lRMSE between the time average of a 10-year ground truth from PlaSim and a 10-year emulation for each 3D variable.

For PDE tasks, the statistical consistency of fluid flows is evaluated by comparing the power spectrum of predicted and true rollouts with spectral RMSE (SRMSE). Exact predictions of turbulent flows over time is usually not possible for PDE surrogates, therefore spectral metrics can quantify the distribution and scale of predicted features rather than relying on point-wise accuracy. Following Rozet et al. (2025), the power spectrum is calculated and partitioned into three evenly distributed frequency bands and reported as the RMSE of the relative power spectrum ($\sqrt{(1 - p/p_\theta)^2}$). For weather forecasting, the statistical performance of probabilistic predictions is measured using the continuous ranked probability score (CRPS) and the spread-skill ratio (SSR). CRPS is minimized when samples from the generative model are drawn from the same distribution as the data. SSR values of 1 are considered optimal as the uncertainty of the forecast matches its error (Fortin et al., 2014). Additional information on the considered metrics is given in the Appendix E.2.

## 4 RESULTS

**Understanding Distances for Physical Distributions** A major hypothesis of this work is that the distributions of current and future states are closer together than Gaussian noise and future states, which allows stochastic interpolants to be more efficient or accurate than conditional generation. Consider an example where initial states are uniformly sampled from a set of initial conditions. For dissipative PDEs, this uniform distribution is transported over time to a stationary distribution as

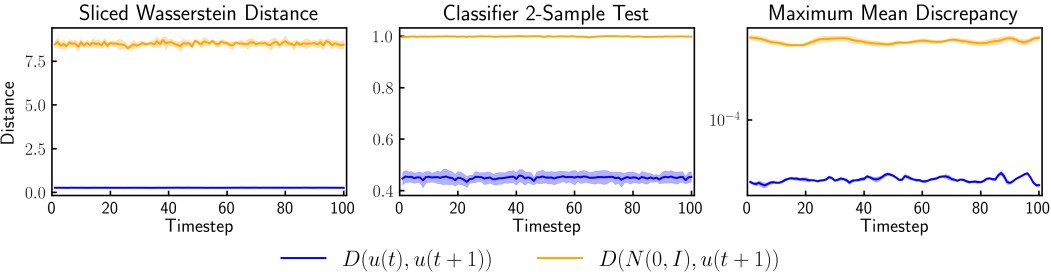

Figure 2: Distance heuristics for the Kolmogorov Flow dataset. Distances between samples drawn from successive timesteps $D(u(t), u(t + 1))$ and between Gaussian noise and future timesteps $D(N(0, I), u(t + 1))$ are plotted for the vorticity field. Each metric is averaged over a 5-fold cross validation, with the standard deviation shaded.

energy is lost. If the time interval is small, we may intuitively believe that subsequent distributions are close; however, we seek to visualize and loosely quantify this difference.

Calculating statistical distances between high-dimensional empirical distributions is usually intractable. Despite this, there are several heuristics that are used. For example, Fréchet Inception Distance (FID) (Heusel et al., 2018) computes distances based on activations of a neural network; while this has no mathematical basis, it is useful to estimate distances between image distributions. We consider more general heuristics such as the Sliced Wasserstein Distance (SW), Classifier 2-Sample Test (C2ST), and Maximum Mean Discrepancy (MMD) (Bischoff et al., 2024). For example, C2ST trains a classifier to discriminate samples drawn from two distributions; if the classifier is perfect (100% accuracy), then the distributions can be viewed as farther than if the classifier cannot identify samples (50% accuracy). Additional information on heuristics are given in Appendix E.3.

We adopt a perspective where each timestep $\mathbf{u}_t \in \mathbb{R}^{n_x \times n_y}$ of a dataset is sampled from a different underlying distribution $\mathbf{u}_t \sim \rho_t$. While $\rho_t$ is unknown, we have access to an empirical distribution with $n$ samples $\rho_t^n = \{\mathbf{u}_t^1, \dots \mathbf{u}_t^n\}$, where $n$ is the dataset size. Heuristics are computed using these empirical distributions as well as $n$ samples $\mathbf{z} \in \mathbb{R}^{n_x \times n_y}$ drawn from a Gaussian $\mathbf{z} \sim \mathcal{N}(0, \mathbf{I})$. For the Kolmogorov Flow dataset, we calculate each heuristic $D$ over time for either $D(\rho_t^n, \rho_{t+1}^n)$ or $D(\mathcal{N}(0, \mathbf{I}), \rho_{t+1}^n)$. Results are plotted and shown in Figure 13. Results for the RBC dataset can be found in Appendix E.3 as well.

In general, the heuristics suggest that distances between subsequent timesteps are closer than distances between future timesteps and Gaussian noise. This is further corroborated by physical intuition, where Kolmogorov flows are known to reach a stationary distribution (Musacchio & Boffetta, 2014) due to its periodic boundary conditions and forcing. Therefore, in these scenarios, it may be beneficial to train interpolants rather than generative models based on Gaussian denoising. However, there are many confounding factors. Firstly, distributions that are far by a certain metric (KL-Divergence, FID, C2ST, etc.) does not mean that their transport map is difficult to learn, such as two Gaussians offset by an arbitrarily large mean. Furthermore, modern deep learning has made learning complex transport maps feasible, to the point that differences between a Gaussian and a well-designed source distribution may not matter in practice.

**Kolmogorov Flow** We report the pointwise and spectral performance of models in Table 1, where the lowest errors are shaded and the second-lowest errors are lightly shaded. All metrics are reported as the average of three seeds, and standard deviations are reported in Appendix C. The number of function evaluations (NFEs) needed for a single prediction is shown, and autoencoder (AE) performance is given as a reference. Models are grouped into related training frameworks and EDM and SI models are evaluated with an Euler (-E) or Euler-Maruyama (-EM) sampler.

FNO performs well in Kolmogorov Flow, likely due to the low Reynolds number ($\sim 10^2$) and the consistent spectrum over time due to the sinusoidal forcing. Truncated sampling based on Tweedie's formula does not seem to work well, which suggests iterative sampling is still necessary for generative models. DDIM/DDPM models also under-perform, with high VRMSE and SRMSE metrics, yet EDM performance suggests that this is an issue of parameterizing the forward/reverse process

| Model: | FNO | DDPM | DDIM | TSM | EDM-E | EDM-EM | FM | SI-E | SI-EM | AE |
|---|---|---|---|---|---|---|---|---|---|---|
| NFEs: | 1 | 100 | 10 | 1 | 10 | 20 | 5 | 5 | 20 | 1 |
| VRMSE | 0.621 | 0.684 | 0.735 | 0.835 | 0.616 | 0.620 | 0.557 | 0.567 | 0.569 | 0.011 |
| SRMSE$_{low}$ | 0.064 | 0.078 | 0.335 | 0.124 | 0.063 | 0.068 | 0.070 | 0.058 | 0.064 | 0.011 |
| SRMSE$_{mid}$ | 0.042 | 0.053 | 0.180 | 0.068 | 0.043 | 0.044 | 0.042 | 0.041 | 0.045 | 0.006 |
| SRMSE$_{high}$ | 0.380 | 0.795 | 0.801 | 0.850 | 0.792 | 0.792 | 0.791 | 0.792 | 0.795 | 0.791 |

Table 1: Pointwise (VRMSE) and Spectral (SRMSE) errors of models on Kolmogorov Flow.

| Model: | FNO | DDPM | DDIM | TSM | EDM-E | EDM-EM | FM | SI-E | SI-EM | AE |
|---|---|---|---|---|---|---|---|---|---|---|
| NFEs: | 1 | 100 | 10 | 10 | 10 | 50 | 5 | 5 | 50 | 1 |
| VRMSE | >10 | 0.765 | 0.675 | 8.961 | 0.681 | 0.699 | 0.733 | 0.637 | 0.726 | 0.027 |
| SRMSE$_{low}$ | 0.357 | 0.405 | 0.612 | 1.113 | 0.363 | 0.329 | 0.323 | 0.449 | 0.296 | 0.086 |
| SRMSE$_{mid}$ | 1.739 | 0.242 | 0.644 | 0.883 | 0.321 | 0.261 | 0.243 | 1.237 | 0.184 | 0.061 |
| SRMSE$_{high}$ | 2.406 | 1.822 | 3.276 | 1.133 | 2.594 | 2.194 | 2.478 | 8.014 | 2.096 | 1.528 |

Table 2: Pointwise (VRMSE) and Spectral (SRMSE) errors of models on Rayleigh-Bénard.

rather than diffusion itself. Further simplifying the stochastic processes to linear interpolation also provides benefits, as flow matching results in the lowest VRMSE. Stochastic interpolants are able to match this pointwise error while having slightly better spectral performance. This is perhaps expected, as FM and SI-E models only differ by the choice of source distribution.

In the relatively laminar Kolmogorov Flow, switching to an SDE-based sampler does not seem to improve performance despite using more sampling steps. Indeed, for smooth dynamics, ODE-based samplers have been shown to be successful in previous work (Li et al., 2025b), and simpler dynamics perhaps avoids the need for a non-Gaussian source distribution. Additionally, for all generative models, pointwise error is largely driven by autoregressive drift rather than the reconstruction error of the autoencoder. In the highest frequency band, the spectral error of the autoencoder also thresholds the spectral error of latent generative models.

**Rayleigh-Bénard Convection** Rayleigh-Bénard Convection offers a more complex system featuring a transition between laminar and turbulent states. Due to the large size of the dataset, only a single set of experiments was run and errors are reported in Table 2. Again, EDM and stochastic interpolant models sampled using either the Euler (-E) or Euler-Maruyama method (-EM).

The performance of FNO matches previous benchmarks from Ohana et al. (2025), where it is unstable across the trajectory. For generative models, Rayleigh-Bénard convection reveals an interesting trade-off where lower pointwise error usually comes at the cost of higher spectral error. After convective mixing, it is intractable for models to exactly predict turbulent states, especially over 50-100 autoregressive predictions. Therefore, minimizing pointwise error tends to push models toward overly smoothed predictions. This behavior can be mitigated by increasing the sampling length or introducing more stochasticity and taking smaller, random steps adds perturbations to help recover the true spectrum. However, greater stochasticity causes predictions to deviate further from the exact state, even as the spectral characteristics remain consistent. This can be qualitatively seen in Figure 8, and suggests that designing generative models for physical emulation requires understanding the metrics of interest and task at hand, rather than using a generic framework.

Deterministically sampled stochastic interpolants achieve the lowest VRMSE, yet exhibit large spectral errors. This is remedied by sampling the model stochastically, where the interpolant is able to achieve low spectral errors. Other generative models fall somewhere along this spectrum. DDPM, EDM-EM, and FM have good spectral accuracy, while EDM-E and DDIM have better pointwise accuracy. In general, this task is very challenging; pointwise errors are high and no model can resolve the highest frequency band, with SRMSE values above 1 being largely meaningless. Additional plots of VRMSE/SRMSE over time for KM and RBC can be found in Appendix B.

**Weather Forecasting** Many commonly used PDEs are deterministic; when fully observed, a future state should be known provided that a sufficiently small time interval is used. Although un-

| Var: | z500 $[m]$ | | | | t2m $[K]$ | | | | t850 $[K]$ | | | | u250 $[m/s]$ | | | | pr_6h $[mm]$ | | | |
|---|---|---|---|---|---|---|---|---|---|---|---|---|---|---|---|---|---|---|---|---|
| Days: | 1 | 3 | 5 | 10 | 1 | 3 | 5 | 10 | 1 | 3 | 5 | 10 | 1 | 3 | 5 | 10 | 1 | 3 | 5 | 10 |
| SFNO | 11 | 33.5 | 57.1 | 119 | 1.27 | 2.33 | 3.28 | 7.55 | 1.17 | 2.21 | 3.24 | 7.94 | 2.37 | 5.26 | 7.99 | 16.0 | 0.87 | 1.29 | 1.47 | 2.27 |
| DDPM | 8.5 | 18.2 | 30.9 | 61.9 | 0.93 | 1.38 | 1.88 | 3.03 | 0.95 | 1.40 | 1.98 | 3.27 | 1.85 | 3.32 | 4.88 | 8.44 | 0.85 | 1.14 | 1.33 | 1.57 |
| DDIM | 7.3 | 15.6 | 26.5 | 54.9 | 0.84 | 1.24 | 1.67 | 2.76 | 0.88 | 1.26 | 1.75 | 2.99 | 1.65 | 2.89 | 4.20 | 7.44 | 0.78 | 1.01 | 1.18 | 1.40 |
| EDM-E | 7.2 | 15.4 | 26.6 | 57.6 | 0.83 | 1.21 | 1.65 | 2.87 | 0.87 | 1.24 | 1.76 | 3.13 | 1.64 | 2.88 | 4.21 | 7.71 | 0.73 | 1.00 | 1.21 | 1.50 |
| FM | 7.1 | 15.2 | 26.5 | 57.7 | 0.79 | 1.15 | 1.60 | 2.81 | 0.84 | 1.19 | 1.72 | 3.07 | 1.60 | 2.83 | 4.18 | 7.82 | 0.72 | 0.98 | 1.19 | 1.49 |
| SI-E | 6.6 | 14.6 | 25.1 | 55.1 | 0.75 | 1.14 | 1.58 | 2.81 | 0.82 | 1.18 | 1.67 | 3.01 | 1.52 | 2.69 | 3.95 | 7.45 | 0.63 | 0.93 | 1.16 | 1.49 |

Table 3: lRMSE of benchmarked models on weather forecasting. Errors are reported at different lead times ($\{1, 3, 5, 10\}$ days) and climate variables. DDPM uses 100 sampling steps, while EDM, DDIM, and SI use 10 steps. FM uses 5 sampling steps.

certainty and statistical metrics are useful for PDEs, climate systems benefit more directly from probabilistic modeling due to the need for well-calibrated forecasts as well as inherent uncertainty in weather data collection and numerical weather prediction (Palmer, 2019; Dueben et al., 2022; Bracco et al., 2025). This makes weather forecasting a good benchmark not only for evaluating the accuracy of generative models but also for understanding their ability to approximate underlying distributions and to capture uncertainty.

After training, models are evaluated on medium-range weather forecasting for up to 10 days. The latitude-weighted RMSE (lRMSE) of models is reported in Table 3. At this time horizon, we observe that ODE-based methods tend to do well. One hypothesis is that, while complex, global weather systems evolve at multiple timescales, including low-frequency variability, which increases the large-scale predictability compared to turbulence or other chaotic PDEs. In this scenario, subsequent states separated by 6 hours tend to still be related, which stochastic interpolants can leverage during sampling. Not only does this produce more accurate forecasts, using 10 sampling steps also pushes the frontiers of efficiency for generative weather models, where it is currently typical to use around 20-40 steps Price et al. (2024); Couairon et al. (2024); Zhuang & Duraisamy (2025).

Within this timescale, we evaluate the probabilistic performance of generative models by calculating the CRPS and SSR of ensemble forecasts. Ensembles are initialized at every 3rd day in the validation year to make a 30-day forecast; the CRPS/SSR is calculated at each timestep and metrics are averaged across all initializations. The resulting CRPS and SSR plots are shown in Figure 3. DDPM is omitted due to its computational expense and its performance on 10-day forecasts. To generate different ensemble members, stochastic interpolants are sampled by using the Euler-Maruyama method (-EM), which is run with 10 steps.

Up to 30 days, stochastic interpolants shows lower CRPS and better-calibrated SSR values. Up to a constant, the CRPS is equal to the squared L2 error between the true and predicted cumulative distribution functions (Zamo & Naveau, 2018), suggesting that stochastic interpolants have a lower distributional shift over time. While most generative models are under-dispersive at shorter lead times, stochastic interpolants tend to mitigate this and calibrate their uncertainty earlier. Additionally, the uncertainty of stochastic interpolants can be tuned by adjusting the noise coefficient of the EM sampler, which allows the model to produce ensembles with more or less variance.

**Climate Emulation** An advantage of using PlaSim instead of real-world data is the availability of a very large dataset of true samples from a long PlaSim integration. This allows models to be evaluated for long-term climate consistency. After training to predict states at 6-hour intervals, generative models are queried to make emulations up to 10 years. DDIM, EDM, and SI-EM are run with 10 steps, and FM is run with 5 steps. For each variable

| Var: | z500 | t2m | t850 | u250 | pr_6h | hus850 |
|---|---|---|---|---|---|---|
| DDIM | 15.7 | 0.55 | 0.45 | 1.91 | 0.125 | 0.437 |
| EDM-E | 9.66 | 0.54 | 0.56 | 1.34 | 0.148 | 0.380 |
| FM | 8.06 | 0.35 | 0.31 | 1.30 | 0.081 | 0.268 |
| SI-EM | 10.7 | 0.51 | 0.48 | 1.03 | 0.069 | 0.187 |

Table 4: 10-year Climatological Biases.

and at each grid point and pressure level, predictions are averaged across all timesteps; the lRMSE between true and predicted averages is the 10-year climatological bias. Prior work has observed the lack of correlation between medium-range forecast error and climatological biases, stemming from error accumulation as the model trained for fast, weather dynamics is integrated to climate

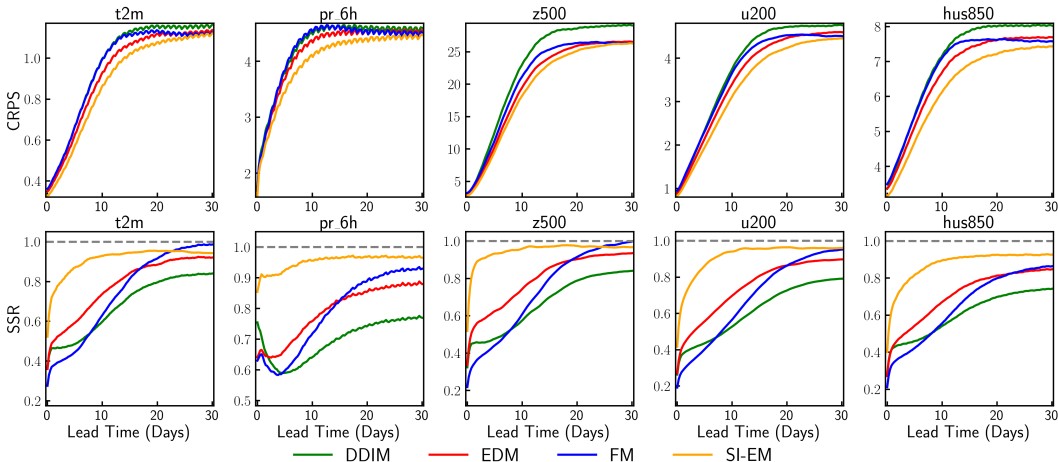

Figure 3: CRPS and SSR plots for the considered generative models over 30 days.

(Chattopadhyay & Hassanzadeh, 2023; Cachay et al., 2024; Watt-Meyer et al., 2025). Therefore, while 10-day forecasting errors converge and are reported after 30 epochs, models are fine-tuned for an additional 20 epochs in order for their climatological biases to also converge. We do not use unrolled training or other specialized methods, instead we simply calculate the 10-year bias at each epoch and the model with the lowest bias is chosen and reported in Table 4.

In general, learning a consistent, long-term climate emulator with uniformly small biases globally from 6-hour prediction intervals is challenging. No model is the best over all variables, and differences between models and from epoch to epoch are large. Smoother fields such as temperature or geopotential are usually better modeled using deterministic samplers, while higher frequency fields such as wind speed, precipitation, or humidity are better modeled with stochastic samplers. Random perturbations added by the EM sampler may help to resolve high-frequency features but can lead to inconsistent trends in smoother fields. This can allow SI models to use different samplers based on the variable of interest, which can be done without re-training.

At longer time horizons of up to 100 years, we track global temperature and precipitation to visualize trends in long-term model performance, shown in Figure 4. In general, most models respond well to the forcing of the seasonal cycle, leading to globally averaged timeseries that are dominated by a periodic, annual timescale. Although models occasionally underestimate or overestimate global temperature, this effect is more consistent in more challenging fields such as precipitation. Despite this, interpolants have the ability to match or exceed other models in matching long-term trends, depending on how they are sampled. Future work can perhaps mitigate model biases or find better training strategies to ensure long-term consistency.

## 5 DISCUSSION

Throughout the work, stochastic interpolants have shown promise as a generative model for modeling physical systems. The key inductive bias that is leveraged is the assumption that successive states are related over a given time interval. When architectures and parameterizations are kept constant, empirical evidence suggests that this is an effective way to train and sample a generative model. However, when the system becomes too simple or models are rolled out for extremely long horizons, assumptions of coupled source and target distributions become less helpful. No single generative model outperforms others on all evaluations (i.e., simple vs. complex systems, pointwise vs. spectral accuracy, etc.) which suggests that current training and sampling frameworks should be tailored based on the problem at hand.

Beyond interpolants, there are interesting trends in other generative models. EDM and flow matching are remarkably consistent in their improvement over DDPM, suggesting that reparameterizing vanilla diffusion with an ODE sampler offers benefits in accuracy and speed. Conditional distribu-

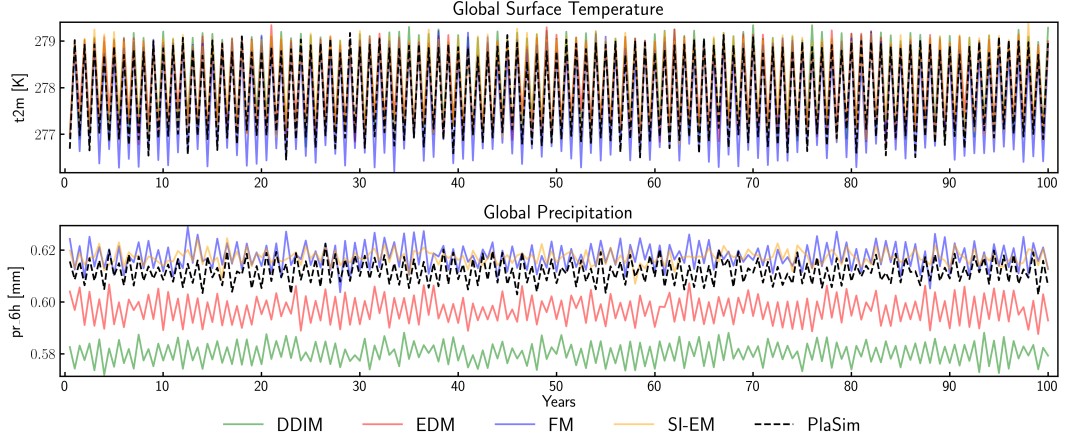

Figure 4: Global surface temperature and precipitation averaged for every 6 months over 100 years.

tions in physical systems tend to be unimodal, as a single state is usually the most probable when observing a past state. Therefore, adding noise in the sampling process when solving the reverse SDE may not be necessary to mix different modes. However, unimodality can be violated in highly chaotic systems, which can result in worse spectral performance when using an ODE sampler.

Finally, to better understand the benchmarked generative models, we provide a set of ablation studies in Appendix C. Consistent with prior evidence (Rozet et al., 2025) we show that the compression ratio of the autoencoder does not influence prediction error substantially. We also corroborate prior results (Li et al., 2025b) that probabilistic training is more effective than deterministic neural solvers in latent space. We additionally verify that generative models are more effective in latent space than in pixel space. Lastly, for stochastic interpolants we investigate the effect of the number of sampling steps and the amount of noise added. Adding noise to the interpolant encourages modes from the source and target distributions to mix, and produces a smoother trajectory through probability space. However, too much noise can transport intermediate distributions too far from the source or target distributions and increase the difficulty of learning a drift or sampling the stochastic process.

## 6 RELATED WORKS

**Generative Models**   There are several foundational works on which generative models and stochastic interpolants are based. Diffusion models (Ho et al., 2020; Song et al., 2021), and its extensions to different samplers (Song et al., 2022; Karras et al., 2022) and flow matching (Lipman et al., 2023; Liu et al., 2022) propose a variety of methods to transport Gaussian noise to an arbitrary density, with the goal of sampling from a target distribution. Subsequent work has relaxed the requirement for a Gaussian prior distribution, allowing transport between arbitrary densities. These models fall under many frameworks, such as Schrödinger Bridges (Bortoli et al., 2023), Diffusion Bridges (Su et al., 2023; Zhou et al., 2023), Optimal Transport (Tong et al., 2024), or Stochastic Interpolants (Albergo et al., 2023).

**Applications to PDEs/Climate**   The use of deep learning to approximate PDE or climate systems is a diverse field, with approaches based on transformers (Pathak et al., 2022; Li et al., 2023; Bi et al., 2022; Li et al., 2024), large-scale pretraining (Zhou & Farimani, 2024; McCabe et al., 2024; Zhou et al., 2024; Nguyen et al., 2023), or physics-based priors (Verma et al., 2024; Zhou & Farimani, 2025). A subset of these works study the application of diffusion models to predict PDE systems (Yang & Sommer, 2023; Huang et al., 2024; Shu et al., 2023; Cachay et al., 2023; Du et al., 2024; Serrano et al., 2024; Molinaro et al., 2025; Gao et al., 2024). Extensions of diffusion models have also been investigated to introduce physics-informed losses (Bastek et al., 2025), operate on meshes (Lino et al., 2025), or use text-conditioned generation (Zhou et al., 2025). Recent progress in flow matching has also been reflected in emulating PDEs, with works that report additional speed or accuracy benefits (Baldan et al., 2025; Utkarsh et al., 2025; Armegioiu et al., 2025; Shi et al., 2024).

Similar applications are also prevalent in climate prediction, where diffusion and flow matching models are popular approaches (Zhuang & Duraisamy, 2025; Cachay et al., 2024; Couairon et al., 2024). Beyond prediction, climate downscaling and data assimilation are also relevant applications of diffusion models (Mardani et al., 2024; Gong et al., 2024; Gao et al., 2023; Andry et al., 2025; Aich et al., 2024; Tomasi et al., 2025; Brenowitz et al., 2025).

Interpolant or diffusion-bridge approaches in physical systems are less common. Previous works have addressed super-resolution and data assimilation in PDEs and climate (Bischoff & Deck, 2023; Schiødt et al., 2025; Chen et al., 2025b; Rout et al., 2025), motivated by their resemblance to denoising/inpainting tasks where stochastic interpolants have proven effective in computer vision. Chen et al. (2024) applies interpolants to forecast stochastic PDEs, and Mücke & Sanderse (2025) show that stochastic interpolants can perform well in modeling fluid problems. We build on these prior works to perform a benchmark study, similar to prior work benchmarking diffusion models (Shysheya et al., 2024; Kohl et al., 2024; Rozet et al., 2025) to better understand and motivate the application of stochastic interpolants to physical emulation. We hope that our experiments can provide further insight into when stochastic interpolants perform well and why these capabilities are observed. As a whole, we show that stochastic interpolants can be a strong baseline for modeling PDE and climate systems, while also providing insights into a variety of different generative models.

## 7    CONCLUSION

**Limitations**    There is ample room to continue investigating stochastic interpolants within physical emulation. Being a very general framework, it can accommodate different source distributions beyond the Gaussian and empirical distributions considered in this work. In fact, these distributions can be seen as the limits of a continuum of source distributions defined as partially noised current states. Much like EDM (Karras et al., 2022), there are many parameters to change with the source distribution, training loss, noise schedule, and sampling framework and studying these can potentially result in further performance gains for stochastic interpolants.

**Takeaways**    Modeling physical phenomena is challenging, as each system exhibits distinct dynamics, variables, and spatial or temporal scales. Although generative models have shown promise for such tasks, their performance varies greatly between the choice of source distribution and stochastic process. Even with the same architecture and training budget, changes in the loss objective and sampling can result in different deterministic and statistical performance. Despite this, stochastic interpolants can be a good baseline for generative models, motivated by the proximity of subsequent states in autoregressive prediction. We hope future work can continue to investigate this and expand the capabilities of generative models for physical systems, such as improving performance in turbulent systems or forecasting weather extremes (Sun et al., 2025).

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

# A  DERIVATION OF AN INTERPOLANT ODE

According to Chen et al. (2024), the conditional drift $b_\theta(t, x_t, x_0)$ for the stochastic interpolant:

$$x_t = \alpha(t)x_0 + \beta(t)x_1 + \sigma(t)W_t \tag{5}$$

can be learned by minimizing the loss:

$$\mathcal{L}_b[b_\theta] = \frac{1}{N}\sum_{i=1}^{N}\left(|b_\theta(t_i, x_{t_i}^i, x_0^i) - (\dot{\alpha}(t)x_0^i + \dot{\beta}(t)x_1^i + \dot{\sigma}(t)W_t)|^2\right) \tag{6}$$

Samples from the conditional probability density $p(x_t|x_0)$ can be generated by solving the SDE:

$$dx_t^{SDE} = b_\theta(t, x_t, x_0)dt + \sigma(t)dW_t \tag{7}$$

This is identical to the setup considered in Section 2 in our work. We consider a case where we would like to sample an ODE such that the generated samples can come from the same distribution as the underlying interpolant and its associated SDE. It will become important later to have values of $\alpha(t), \beta(t), \sigma(t)$, which we have set to $\alpha(t) = 1 - t, \beta(t) = t, \sigma(t) = 1 - t$ in this work (also consistent with Section 2).

Again, from Chen et al. (2024), the learned drift $b_\theta(t, x_t, x_0)$ has an associated SDE (Equation 7), but can also be manipulated to define a family of SDEs:

**Theorem A.1 (Families of SDEs (Chen et al. (2024), Theorem 3.2))** *Given any $g \in C^0[0, 1]$ such that $\lim_{t\to 0^+} \frac{g(t)^2 - \sigma(t)^2}{t}$ and $\lim_{t\to 1^-} \frac{g(t)^2}{\sigma(t)}$ exists, define:*

$$\hat{b}_\theta(t, x_t, x_0) = b_\theta(t, x_t, x_0) + \frac{1}{2}(g_t^2 - \sigma_t^2)\nabla \log p(x_t|x_0) \tag{8}$$

*where $b_\theta(t, x_t, x_0)$ is the minimizer of the loss in Equation 6. Then the solutions to the SDE:*

$$d\hat{x}_t^{SDE} = \hat{b}_\theta(t, x_t, x_0)dt + g(t)dW_t \tag{9}$$

*are such that $Law(\hat{x}_t^{SDE}) = Law(x_t^{SDE}) = Law(x_t|x_0)$ for all $t, x_0$.*

Interestingly, we can set $g(t) = 0$ to change Equation 9 into an ODE:

$$d\hat{x}_t^{ODE} = \hat{b}_\theta(t, x_t, x_0)dt \tag{10}$$

One of the regularity conditions is not met by this choice of $g(t)$ ($\lim_{t\to 0^+} \frac{g(t)^2 - \sigma(t)^2}{t}$), however, we will see later how this can be remedied numerically by our particular choice of $\alpha(t), \beta(t), \sigma(t)$. For now, what remains is to derive the modified drift $\hat{b}_\theta(t, x_t, x_0)$ in Equation 8 by calculating the score function $\nabla \log p(x_t|x_0)$. From Chen et al. (2024), the score function can be expressed as:

$$\nabla \log p(x_t|x_0) = A_t(\beta(t)b_\theta(t, x_t, x_0) - c_t(x_t, x_0)) \tag{11}$$

where

$$A_t = \frac{1}{t\sigma(t)(\dot{\beta}(t)\sigma(t) - \beta(t)\dot{\sigma}(t))}, \quad c_t(x_t, x_0) = \dot{\beta}(t)x_t + (\beta(t)\dot{\alpha}(t) - \dot{\beta}(t)\alpha(t))x_0 \tag{12}$$

Substituting in our choices for $\alpha(t) = 1 - t, \beta(t) = t, \sigma(t) = 1 - t$ results in:

$$A_t = \frac{1}{t(1 - t)}, \quad c_t(x_t, x_0) = x_t - x_0 \tag{13}$$

Therefore the score function becomes:

$$\nabla \log p(x_t|x_0) = \frac{1}{t(1 - t)}(tb_\theta(t, x_t, x_0) - (x_t - x_0)) \tag{14}$$

and the drift for the ODE in Equation 10 becomes:

$$\hat{b}_\theta(t, x_t, x_0) = \frac{1 + t}{2}b_\theta(t, x_t, x_0) + \frac{1 - t}{2t}(x_t - x_0) \tag{15}$$

The regularity condition $\lim_{t \to 0^+} \frac{g(t)^2 - \sigma(t)^2}{t}$ exists in order to make the modified drift $\hat{b}_\theta(t, x_t, x_0)$ exist at $t = 0$ for all choices of $\alpha(t), \beta(t), \sigma(t)$ and $x$ (Chen et al. (2024), Appendix B.4). However, given our coefficients, we can see that during sampling at $t = 0$, $x_t - x_0 = 0$ and therefore:

$$\hat{b}_\theta(0, x_t, x_0) = \frac{1}{2} b_\theta(0, x_t, x_0) \tag{16}$$

This expression can be analytically solved during the first integration step at $t = 0$ to avoid singularities in Equation 15. Additionally, scaling the noise coefficient with $\sigma(t) = \sigma(1 - t)$ does not change the modified drift, as the scaling term $\sigma$ eventually cancels out.

To understand this modified drift, we consider a hypothetical scenario. In the ideal case, $x_t$ should equal the interpolant in Equation 5 when integrating the ODE (although this is never true due to approximation errors in $b_\theta(t, x_t, x_0)$). Substituting for $x_t$ in Equation 15:

$$\hat{b}_\theta(t, x_t, x_0) = \frac{1+t}{2} b_\theta(t, x_t, x_0) + \frac{1-t}{2} \left( (x_1 - x_0) + \frac{(1-t)}{t} W_t \right) \tag{17}$$

When the loss (Equation 6) is minimized, the learned drift becomes $b_\theta(t, x_t, x_0) = (x_1 - x_0) - W_t$, and therefore the adjusted drift becomes:

$$\hat{b}_\theta(t, x_t, x_0) = (x_1 - x_0) + \left( \frac{1 - 3t}{2t} \right) W_t \tag{18}$$

which is equal to the original drift, but with a modified noise term. Interestingly, at $t = 1$, we recover the original drift, and at $t = 1/3$, we recover the deterministic drift: $x_1 - x_0$.

One remark is that while this ODE allows us to numerically approximate a sample from $p(x_1|x_0)$, it does not result in a generative model. Specifically, the ODE cannot produce different samples $x_1$ since it deterministically integrates from $x_0$. The diffusion term in the SDE allows transport from a Dirac delta $p(x_0)$ to a distribution $p(x_1|x_0)$, while the ODE represents the limiting case when the diffusion term approaches zero. Despite this shortcoming, for deterministic prediction, this method still produces results with low errors. Empirically, we have also observed behavior in which the original drift $b_\theta(t, x_t, x_0)$ can be integrated with an ODE without modification and still produce samples with low RMSE-based errors. The reasons to why this behavior occurs is still unknown and an area for future work.

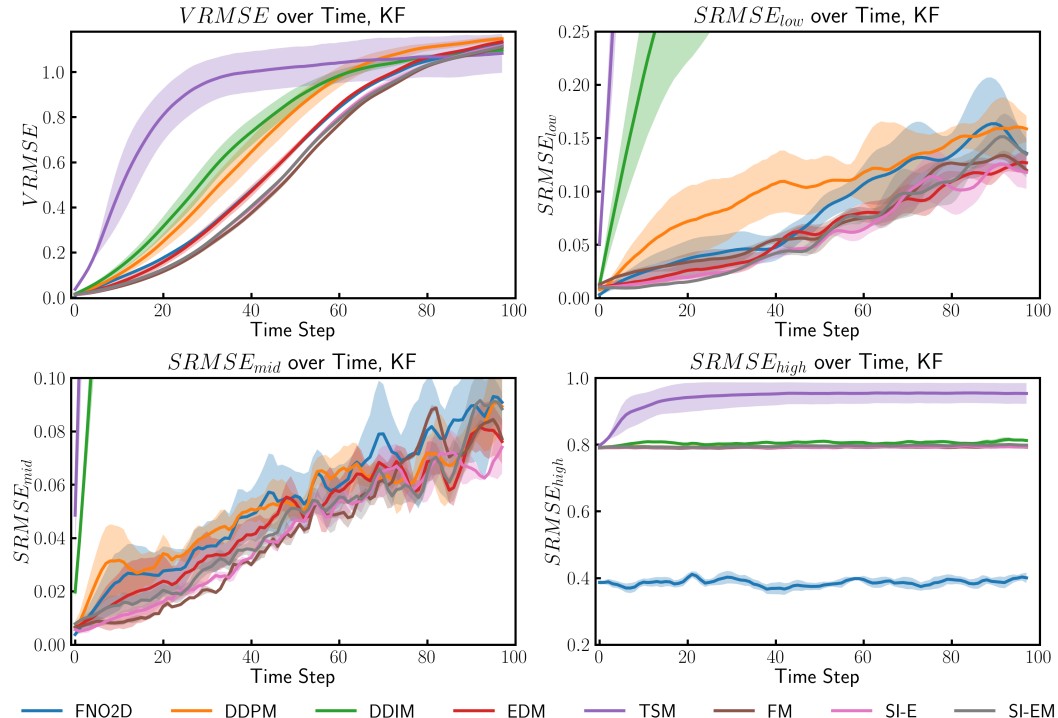

Figure 5: VRMSE/SRMSE for model predictions over time on the Kolmogorov Flow (KF) dataset. Each model is trained with three seeds, mean errors are plotted with standard deviations shaded.

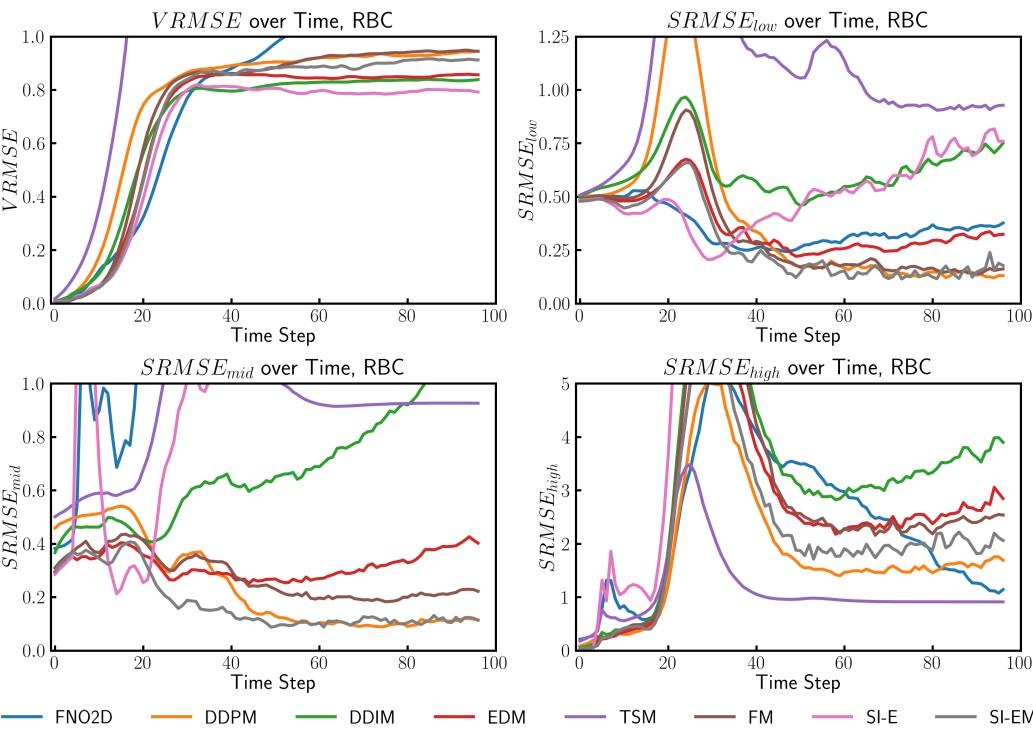

Figure 6: VRMSE/SRMSE for model predictions over time on the Rayleigh-Bénard Convection (RBC) dataset. In general, there is an increase in errors at $\sim t = 20$ as fluid layers mix.

# B  SUPPLEMENTARY RESULTS

## B.1  ERROR PLOTS

To visualize time-dependent trends in model errors, we plot the VRMSE and SRMSE over time for model predictions, averaged over all samples the validation set. Errors for Kolmogorov Flow are plotted in Figure 5 and errors for Rayleigh-Bénard Convection are plotted in Figure 6.

Error trends for Kolmogorov Flow tend to be more clear. Stochastic interpolants and flow matching models have lower VRMSE across the prediction horizon, and all models accumulate error near the middle of the trajectory. When autoregressive drift sufficiently shifts the input distribution, VRMSE growth tends to taper off as predictions reach a steady-state. At the low- and mid-frequency bands, stochastic interpolants also tend to have lower SRMSE across the trajectory. The mid-band SRMSE tends to oscillate more, as higher frequency features tend to form and dissipate more quickly than lower frequency features in Kolmogorov Flow. Lastly, at the high-frequency band, the performance of generative models is thresholded by the SRMSE of the autoencoder ($\sim 0.8$), while FNO can have a lower SRMSE. In general, stochastic interpolants, flow matching, and EDM tend to perform well on this dataset, with good spectral and pointwise accuracy.

Examining time-dependent errors for RBC can also reveal insights. All models rapidly accumulate pointwise errors during turbulent mixing ($\sim t = 20$) and reach steady state during subsequent dissipation. In this regime, all models have have decorrelated from the true trajectory and based on our observations, pointwise accuracy tends to be achieved by smoother fields. In the low-frequency band, we see a spike in error as the fluid undergoes mixing. Despite this, most generative models can recover low-frequency features after mixing. Similar observations can also be made for the mid-frequency SRMSE, however, accurately capturing features after mixing becomes more difficult. At the high-frequency band, most models reach errors above 1 after $t = 20$ and stay there.

These plots shed more insight into tradeoffs between point-wise and spectral accuracy in chaotic systems. DDPM has low spectral error yet has high point-wise error; using a different sampler with DDIM achieves low point-wise error but with high spectral errors. We also observe a similar phenomenon when training stochastic interpolants and using either an ODE or SDE based sampler. Perhaps it is still an open question as to what the desired behavior should be in turbulence modeling or if we can train models to accomplish both point-wise and spectral accuracy.

## B.2  VISUALIZATIONS

To qualitatively evaluate model performance, we provide a set of visualizations of model performance on Kolmogorov Flow (KF) and Rayleigh-Bénard Convection (RBC) in Figures 7 and 8. KF predictions are shown at $t = 50$, when model predictions begin to decorrelate. RBC predictions are shown at $t = 18$ and $t = 45$, which is before and during turbulent mixing. In the turbulent regime, the effects of SDE-based samplers is more clear. DDPM and SI-EM are able to capture more high-frequency features, despite having larger pointwise errors.

Additionally, medium-range forecasts using the PlaSim dataset are plotted in Figure 9 for different models, variables, and lead times. At this length scale and time horizon, differences between models are challenging to distinguish, although they exist. To visualize long-horizon consistency, we look at the zonally-averaged power spectrum of model predictions after 100 years, since this lead time is well beyond the limit for pointwise consistency. After being rolled out to 100 years, we plot the zonally-averaged power spectrum of different model predictions as well as the ground truth in Figure 10. At this timescale, the considered generative models seem to produce similar spectra, although their biases may be different (Figure 4). Although no model is the best, the spectrum of generative models remains largely accurate and remains stable over 100 years, which is a promising sign for their capabilities as climate emulators.

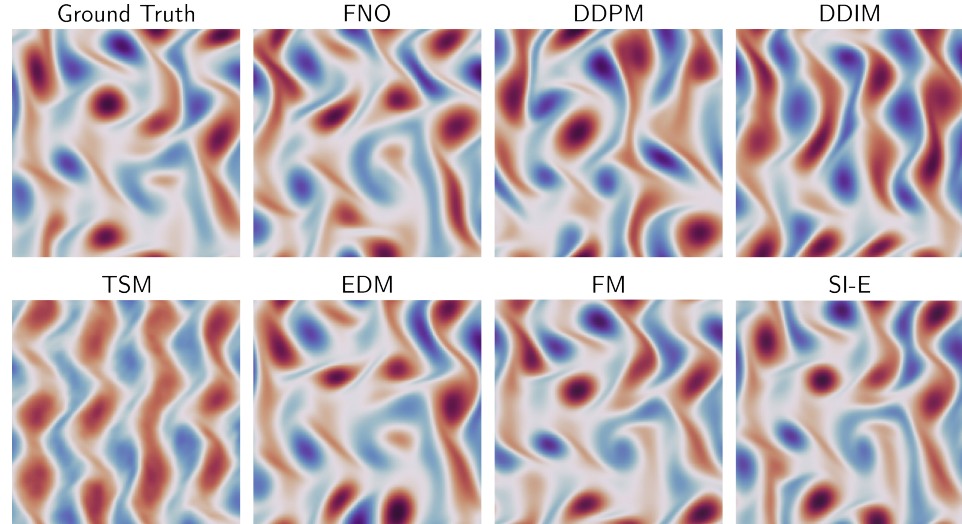

Figure 7: Model predictions of Kolmogorov Flow at $t = 50$, shown with the ground truth. Most models tend to de-correlate at this timestep, although some models better capture relevant features.

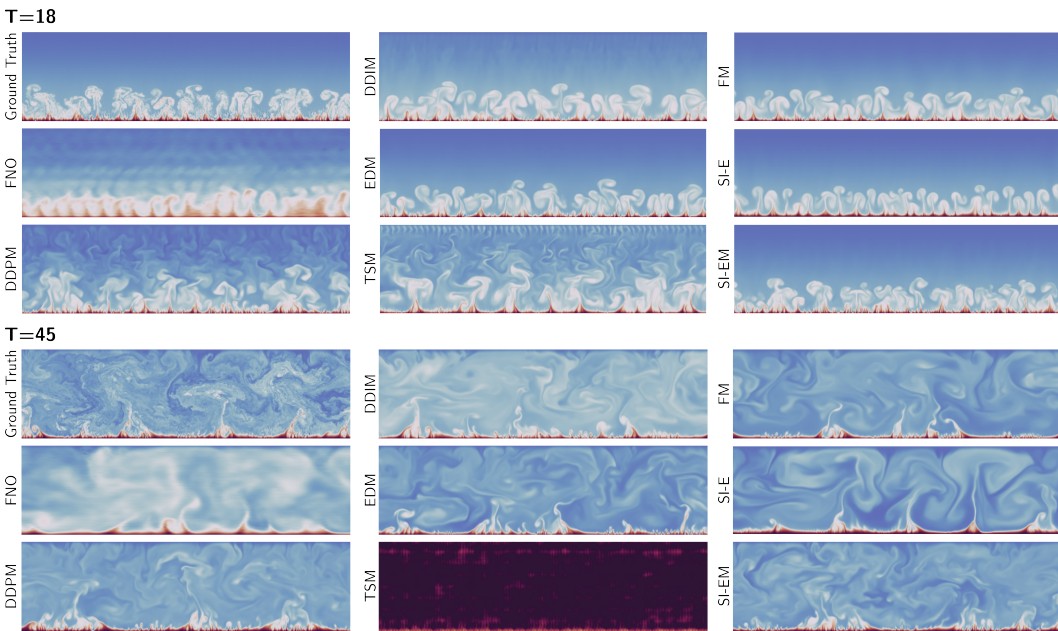

Figure 8: Model predictions of Rayleigh-Bénard Convection at $t = 18$ and $t = 45$, shown with the ground truth. In the laminar regime, most models can model initial mixing. After mixing, the effects of SDE-based samplers become more clear. DDIM/EDM are noticeably smoother than DDPM, likewise, FM/SI-E are smoother than SI-EM.

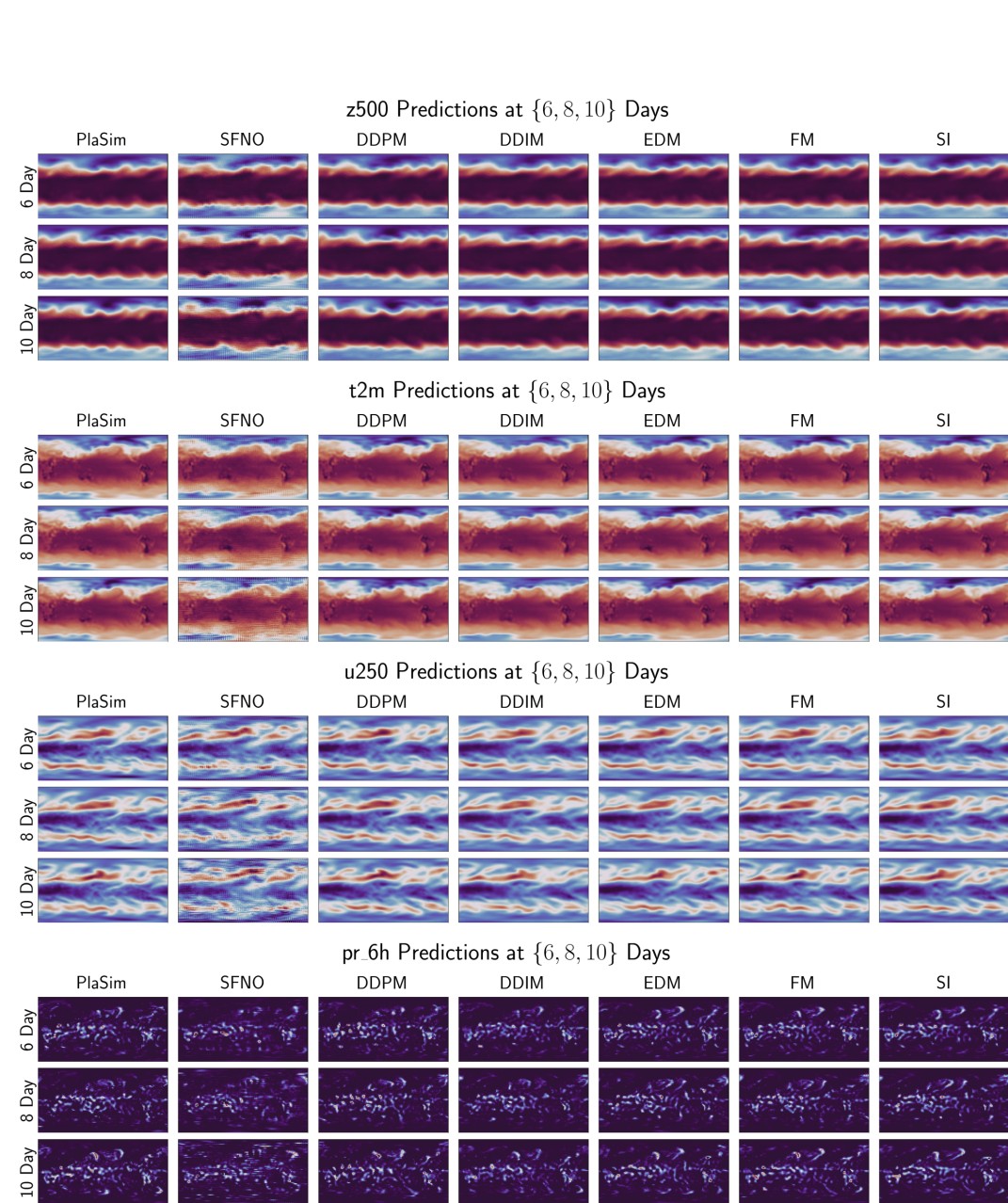

Figure 9: Model predictions of z500, t2m, u250, and pr_6h at lead times of $\{6, 8, 10\}$ days. At a coarse scale $128 \times 64$, latent generative models tend to work well for medium-range weather forecasting.

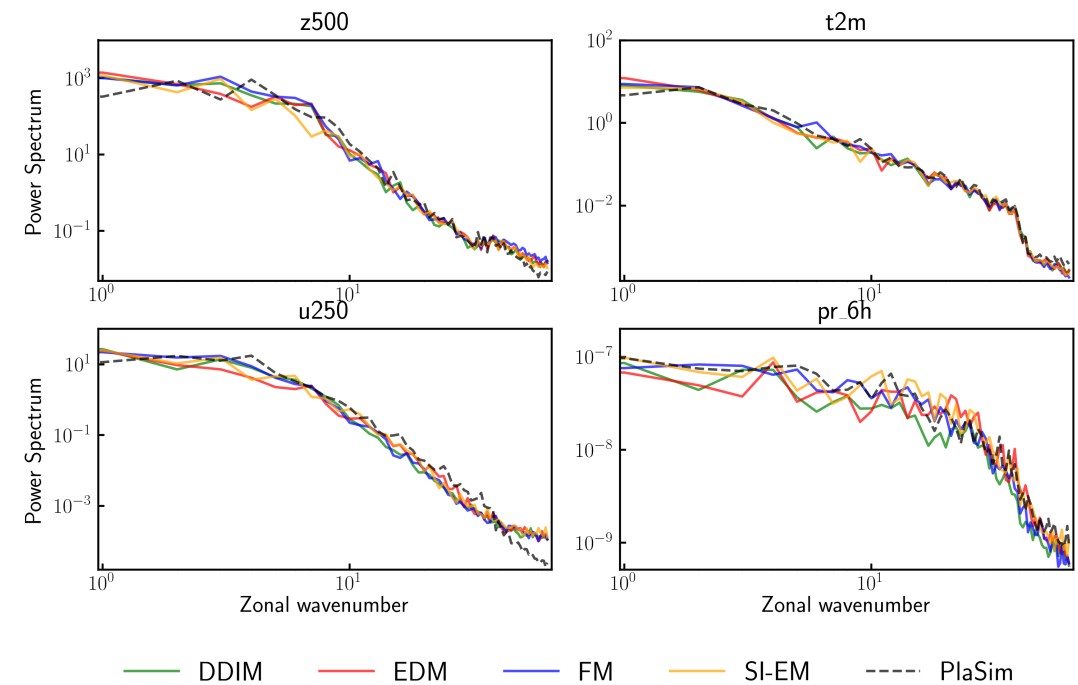

Figure 10: Zonally-averaged power spectra for different models and weather variables for predictions at a 100 year lead time. Despite having different biases, the considered generative models have consistent spectra and remain stable at long horizons.

## C    ABLATION STUDIES

**Pixel/Latent Space Ablations**    We train flow matching and stochastic interpolant models in either pixel or latent space to compare their performance. The Normalized RMSE (NRMSE), or Relative L2 error, is reported for the Kolmogorov Flow validation set in Table 5. Model sizes are kept roughly constant, and all models are trained for the same number of epochs. To account for the larger spatial input to the DiT ($160 \times 160$), a patch size of $8 \times 8$ was used to match the compression ratio of the autoencoder.

In general, we find that pixel space models have higher errors than latent space models, in addition to being more expensive to train and query. One hypothesis is that latent space models could be more stable, as the latent space has less variance and a well-trained decoder may smooth out errors. Indeed, we find that autoregressive drift is the primary contributor to large errors in pixel space models.

| Metric | $\text{FM}_{pixel}$ | $\text{FM}_{latent}$ | $\text{SI}_{pixel}$ | $\text{SI}_{latent}$ |
|---|---|---|---|---|
| NRMSE | 0.812 | 0.649 | 0.966 | 0.609 |

Table 5: Comparison of pixel and latent space generative models on Kolmogorov Flow.

**Compression Ratio Ablations**    We compare the effects of training autoencoders with different compression ratios $(64, 256)$ on the reconstruction error of the autoencoder and the rollout error of the generative model. The NRMSE for Rayleigh-Bénard convection is reported in Table 6. A larger compression ratio results in a smaller latent space, increasing the reconstruction error of the autoencoder. Despite having nearly double the reconstruction error, generative models with more aggressive compression only have a modest increase in rollout error, which is consistent with Rozet et al. (2025).

| $\div$ | AE | FM | SI |
|---|---|---|---|
| 64 | 0.0252 | 0.619 | 0.605 |
| 256 | 0.0457 | 0.649 | 0.622 |

Table 6: Comparison of NRMSE for models using different compression ratios ($\div$) on Rayleigh-Bénard Convection. Reconstruction error is reported for the autoencoder (AE), while rollout error is reported for the generative models.

**Deterministic/Probabilistic Latent Models**  We compare training a deterministic, latent neural solver (LNS) (Li et al., 2025a) against latent generative models in Table 7. LNS is trained to regress future latent states with an MSE loss, rather than a denoising loss. During inference, LNS makes a single future prediction, whereas generative models need to iteratively sample future states. Similar to prior works, we find that probabilistic training is more beneficial than deterministic models in latent space, although the benefit is not as large as previously reported (Rozet et al., 2025).

| Model: | LNS | FM | SI |
|---|---|---|---|
| NFEs: | 1 | 5 | 5 |
| NRMSE | 0.623 | 0.544 | 0.550 |

Table 7: Comparison of latent neural solver (LNS) and generative models on Kolmogorov Flow.

**Number of Sampling Steps**  We compare the effect of using different samplers and numbers of sampling steps for stochastic interpolant models in Table 8. After training on the Kolmogorov Flow dataset, models are either sampled with an Euler sampler or Euler-Maruyama sampler to solve the probability flow ODE or SDE. In general, we find that ODE-based samplers require fewer steps to obtain good performance. Furthermore, performance tends to increase consistently with more sampling steps, although these performance gains will saturate at some point.

| Sampler: | Euler | | | Euler-Maruyama | | |
|---|---|---|---|---|---|---|
| NFEs: | 2 | 5 | 10 | 10 | 20 | 50 |
| NRMSE | 0.560 | 0.548 | 0.537 | 0.555 | 0.535 | 0.534 |

Table 8: Comparison of different samplers and number of sampling steps for SI models on Kolmogorov Flow.

**Noise Coefficients for Interpolants**  The forward and reverse processes for stochastic interpolants are set by Equation 2, or $x(t) = \alpha(t)x_0 + \beta(t)x_1 + \sigma(t)W_t$. We define $\sigma(t) = \sigma(1-t)$ as the noise coefficient, where $\sigma$ controls the scale of the noise in the stochastic process. More noise can encourage mixing between $x_0$ and $x_1$, however, too much noise can make the stochastic process more difficult to learn or sample from. We can see this in Table 9, where SI models are trained to learn stochastic processes with different noise scales $\sigma$ and are sampled with an Euler sampler using 2 steps. In general, $\sigma$ can be tuned to find an optimal amount of noise for the stochastic process.

| $\sigma$: | 0.1 | 0.5 | 1 | 3 |
|---|---|---|---|---|
| NRMSE | 0.702 | 0.632 | 0.609 | 0.803 |

Table 9: Comparison of different $\sigma$ coefficients for SI models in Kolmogorov Flow. $\sigma$ scales the amount of noise in the stochastic process.

**Ablations for DDPM/DDIM/TSM**  DDPM, DDIM, and TSM all model the same forward process, but sample the reverse process differently. We perform a hyperparameter sweep for these baselines in Table 10, and use the best configurations in our experiments. We experiment with different

noise schedules as well as different start and ending points for the noise schedule by modifying a scaling factor. For the linear schedule, we set $\beta_{start} = 0.0001$ and $\beta_{end} = 0.02$ and sweep over scaling factors $S$ that modify the schedule by multiplying $S * \beta_{start}$, $S * \beta_{end}$, following Shehata et al. (2025). Additionally, the cosine noise schedule is modified by changing $s$ in the expression: $\bar{\alpha}_t = \frac{f(t)}{f(0)}, f(t) = cos(\frac{t/T+s}{1+s} * \frac{\pi}{2})^2$. Results are shown in Table 10.

We find that a cosine schedule degrades performance for DDPM and TSM, however DDIM benefits from the cosine schedule, with the scale having a small effect on performance. Within linear schedules, the scale $S = 400$ seems to work well. Therefore we train DDPM, DDIM, and TSM models with a linear schedule with $S = 400$ to allow a comparison between samplers for diffusion training.

| Model | Schedule | Scale | VRMSE | SRMSE$_{low}$ | SRMSE$_{mid}$ | SRMSE$_{high}$ |
|---|---|---|---|---|---|---|
| DDPM | Linear | $S = 200$ | 0.691 | 0.137 | 0.074 | 0.794 |
| DDIM | Linear | $S = 200$ | 0.797 | 1.840 | 0.831 | 0.977 |
| TSM | Linear | $S = 200$ | 0.969 | 1.413 | 2.082 | 0.932 |
| DDPM | Linear | $S = 400$ | 0.655 | 0.085 | 0.056 | 0.794 |
| DDIM | Linear | $S = 400$ | 0.744 | 0.284 | 0.208 | 0.804 |
| TSM | Linear | $S = 400$ | 0.827 | 1.541 | 0.892 | 0.896 |
| DDPM | Linear | $S = 800$ | 0.691 | 0.079 | 0.074 | 0.800 |
| DDIM | Linear | $S = 800$ | 0.671 | 0.080 | 0.098 | 0.799 |
| TSM | Linear | $S = 800$ | >10 | 0.978 | 0.985 | 0.999 |
| DDPM | Cosine | $s = 0.004$ | 7.091 | 0.526 | 0.569 | 0.909 |
| DDIM | Cosine | $s = 0.004$ | 0.677 | 0.276 | 0.187 | 0.801 |
| TSM | Cosine | $s = 0.004$ | >10 | 1.000 | 1.000 | 1.000 |
| DDPM | Cosine | $s = 0.008$ | 7.693 | 0.586 | 0.631 | 0.923 |
| DDIM | Cosine | $s = 0.008$ | 0.679 | 0.266 | 0.171 | 0.800 |
| TSM | Cosine | $s = 0.008$ | >10 | 1.000 | 1.000 | 1.000 |

Table 10: Comparison of hyperparameters for DDPM/DDIM/TSM models in Kolmogorov Flow.

**Ablations for EDM** Elucidated Diffusion Models (EDMs) reparameterize the forward and reverse process to a common form to better optimize their parameters. We experiment with different noise schedules $\sigma_{min}, \sigma_{max}$ as well as sampling parameters $\rho$. Furthermore, we examine an SDE-based sampler by using the Euler-Maruyama method with 20 steps. Following Shehata et al. (2025), we adopt SDE sampling parameters $S_{churn}, S_{min}, S_{max}, S_{noise}$ of $10, 0, 1, 1e6$ based on tuning performed in their work. Results are shown in Table 11.

| Sampler | $\sigma_{min}, \sigma_{max}$ | $\rho$ | VRMSE | SRMSE$_{low}$ | SRMSE$_{mid}$ | SRMSE$_{high}$ |
|---|---|---|---|---|---|---|
| ODE | $0.002, 80$ | 3 | 0.616 | 0.084 | 0.062 | 0.793 |
| ODE | $0.02, 100$ | 3 | 0.682 | 0.236 | 0.061 | 0.792 |
| ODE | $0.001, 60$ | 3 | 0.692 | 0.252 | 0.074 | 0.792 |
| ODE | $0.002, 80$ | 7 | 0.606 | 0.073 | 0.047 | 0.792 |
| ODE | $0.02, 100$ | 7 | 0.685 | 0.194 | 0.059 | 0.791 |
| ODE | $0.001, 60$ | 7 | 0.688 | 0.241 | 0.069 | 0.792 |
| SDE | $0.002, 80$ | 3 | 0.625 | 0.091 | 0.046 | 0.794 |
| SDE | $0.02, 100$ | 3 | 0.677 | 0.208 | 0.054 | 0.792 |
| SDE | $0.001, 60$ | 3 | 0.684 | 0.226 | 0.059 | 0.791 |
| SDE | $0.002, 80$ | 7 | 0.612 | 0.092 | 0.045 | 0.794 |
| SDE | $0.02, 100$ | 7 | 0.680 | 0.204 | 0.053 | 0.791 |
| SDE | $0.001, 60$ | 7 | 0.682 | 0.218 | 0.061 | 0.792 |

Table 11: Comparison of different hyperparameters for EDM models in Kolmogorov Flow.

For simpler systems like Kolmogorov Flow, SDE-based sampling does not seem to help despite using twice the number of sampling steps, except when resolving the mid-band $SRMSE$. Furthermore, the original noise schedule reported in Karras et al. (2022) seems to work the best with

$\sigma_{min} = 0.002, \sigma_{max} = 80$, likely since we normalize our data to a similar range. Lastly, there are small changes in performance due to the weighting of the sampling steps, controlled by $\rho$. Therefore, we opt to use EDM with an ODE sampler and with parameters $\sigma_{min} = 0.002, \sigma_{max} = 80, \rho = 7$.

**Statistical Significance** Training and evaluations for all models in the Kolmogorov Flow dataset were repeated three times across three random seeds. The mean and standard deviation of metrics reported in Table 1 are given in Table 12.

| Model | NFEs | VRMSE | $\text{SRMSE}_{low}$ | $\text{SRMSE}_{mid}$ | $\text{SRMSE}_{high}$ |
|---|---|---|---|---|---|
| FNO | 1 | $0.621_{\pm 0.008}$ | $0.064_{\pm 0.001}$ | $0.042_{\pm 0.001}$ | $0.380_{\pm 0.017}$ |
| DDPM | 100 | $0.684_{\pm 0.022}$ | $0.078_{\pm 0.005}$ | $0.053_{\pm 0.003}$ | $0.795_{\pm 0.001}$ |
| DDIM | 10 | $0.735_{\pm 0.007}$ | $0.335_{\pm 0.073}$ | $0.180_{\pm 0.056}$ | $0.801_{\pm 0.005}$ |
| TSM | 1 | $0.835_{\pm 0.044}$ | $0.124_{\pm 0.029}$ | $0.068_{\pm 0.011}$ | $0.850_{\pm 0.033}$ |
| EDM-E | 10 | $0.616_{\pm 0.013}$ | $0.063_{\pm 0.002}$ | $0.043_{\pm 0.002}$ | $0.792_{\pm 0.001}$ |
| EDM-EM | 20 | $0.620_{\pm 0.005}$ | $0.068_{\pm 0.016}$ | $0.045_{\pm 0.001}$ | $0.793_{\pm 0.001}$ |
| FM | 5 | $0.557_{\pm 0.006}$ | $0.070_{\pm 0.006}$ | $0.042_{\pm 0.001}$ | $0.791_{\pm 0.000}$ |
| SI-E | 5 | $0.567_{\pm 0.002}$ | $0.058_{\pm 0.004}$ | $0.041_{\pm 0.001}$ | $0.792_{\pm 0.000}$ |
| SI-EM | 20 | $0.569_{\pm 0.002}$ | $0.064_{\pm 0.005}$ | $0.045_{\pm 0.002}$ | $0.795_{\pm 0.002}$ |
| AE | 1 | 0.011 | 0.011 | 0.006 | 0.791 |

Table 12: Pointwise (VRMSE) and Spectral (SRMSE) errors of models on Kolmogorov Flow

# D    DATASET INFORMATION

**Kolmogorov Flow** In vorticity form, Kolmogorov Flow can be described by the PDE (Koehler et al., 2024):

$$\frac{\partial \omega}{\partial t} = -b\left(\begin{bmatrix} 1 \\ -1 \end{bmatrix} \odot \nabla(\Delta^{-1}\omega)\right) \cdot \nabla\omega + \nu\nabla \cdot \nabla\omega + \lambda\omega - k\cos(k\frac{2\pi}{L}y) \quad (19)$$

The leftmost term represents vorticity convection, controlled by the coefficient $b$. Diffusion is controlled by the viscosity $\nu$ and a drag term $\lambda\omega$ is introduced. Lastly, a sinusoidal forcing term is introduced, controlled by the magnitude $k$. Coefficients are kept constant throughout data generation. Initial conditions are uniformly sampled from a truncated Fourier series. While not straightforward to write in 2D/3D, in 1D the series is written as:

$$\omega_0 = \sum_{k=1}^{5} a_k \sin(k\frac{2\pi}{L}x + \phi_k) \quad (20)$$

where $L$ is the length of the domain and terms $a_k \in [-1, 1]$ and $\phi_k \in [0, 2\pi]$ are uniformly sampled. This results in a uniform distribution for the initial states of the system, which does not dissipate over time due to the sinusoidal forcing. Furthermore, Kolmogorov flows are well-studied and, after an initial burn-in period, reach a stationary distribution (Musacchio & Boffetta, 2014).

**Rayleigh-Bénard Convection** The equations for Rayleigh-Bénard Convection are governed by a buoyancy and Navier-Stokes equation (Ohana et al., 2025):

$$\frac{\partial b}{\partial t} - \kappa\Delta b = -u \cdot \nabla b \quad (21)$$

$$\frac{\partial u}{\partial t} - \nu u + \nabla p - b\mathbf{e}_z = -u \cdot \nabla u \quad (22)$$

The thermal diffusivity $\kappa$ and viscosity $\nu$ are determined by the Rayleigh and Prandtl numbers:

$$\kappa = (\text{Rayleigh} \times \text{Prandtl})^{-\frac{1}{2}}, \quad \nu = \left(\frac{\text{Rayleigh}}{\text{Prandtl}}\right)^{-\frac{1}{2}} \quad (23)$$

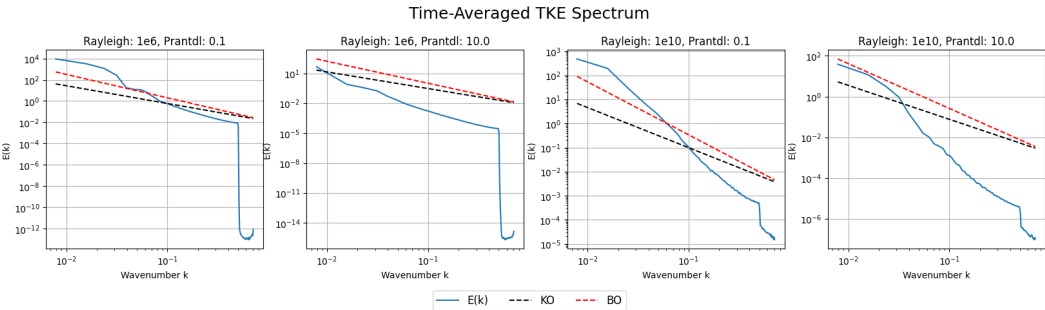

Figure 11: Plots of time-averaged turbulent kinetic energy spectrum at two different Prandtl and Rayleigh numbers. Kinetic energy is averaged after the burn-in period, and both Kolmogorov-Obukhov (KO) and Bolgiano-Obukhov (BO) power laws are plotted.

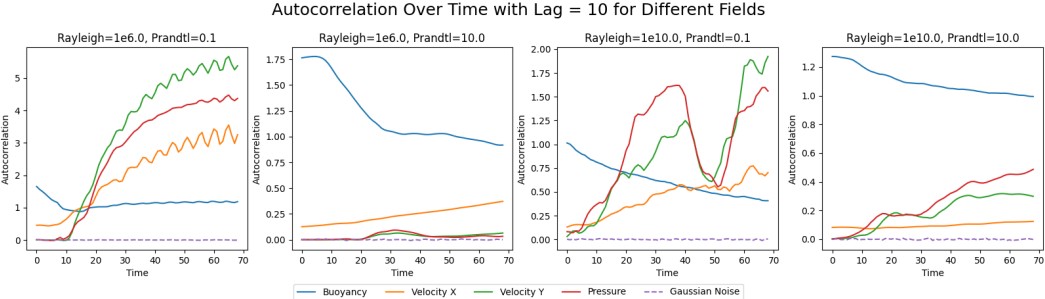

Figure 12: Autocorrelation over time, measured at different Rayleigh and Prandtl numbers and calculated after the burn-in period. The lag $\tau$ is chosen such that increasing $\tau$ does not change the autocorrelation appreciably. Different fields are plotted as well as a Gaussian distribution for reference. Constant autocorrelation denotes convergence to a stationary distribution.

Rayleigh and Prandtl numbers are varied throughout the training and validation data, where the Prandtl number measures the relative importance between momentum diffusivity and thermal diffusivity and the Rayleigh number measures the relative importance between buoyancy forces and viscous forces in heat transfer. In particular, Rayleigh $\in \{1e6, 1e7, 1e8, 1e9, 1e10\}$ and Prandtl $\in \{0.1, 0.2, 0.5, 1, 2, 5, 10\}$. Furthermore, initial conditions for the buoyancy are generated by $b(t = 0) = (Ly - y) \times \delta b_0 + y(Ly - y) \times \epsilon$, where $\delta b_0$ is sampled from $\{0.2, 0.4, 0.6, 0.8, 1.0\}$ and $\epsilon$ is sampled from a Gaussian scaled to $10^{-3}$. This results in a linear buoyancy gradient in the vertical direction with a small perturbation. All other fields are initialized to zero. Therefore, the initial conditions can be approximated by a categorical distribution, but as the system evolves, each sample quickly diverges and produces unique trajectories.

The statistical features of Rayleigh-Bénard Convection are more diverse than Kolmogorov Flow. After mixing, the flows are turbulent, however, their characteristics usually do not follow theoretical trends and vary as Rayleigh and Prandtl numbers can span many orders of magnitude. A common way to study turbulent flows is to look at the energy spectrum of the time-averaged turbulent kinetic energy (TKE), defined as the time average of $\frac{1}{2}((u - \bar{u})^2 + (v - \bar{v})^2)$, where $\bar{u}, \bar{v}$ are time-averaged velocities. We plot the energy spectrum of different Rayleigh-Bénard samples in Figure 11. Additionally, we compare these spectra to theoretical power laws, such as the Kolmogorov-Obukhov law for stationary, homogeneous and isotropic turbulence: $E(k) \sim \epsilon^{2/3} k^{-5/3}$, where the energy dissipation is approximated as $\epsilon = U^3/L$ with velocity magnitude $U$ and length scale $L$. Additionally, prior work has suggested that Rayleigh-Bénard Convection is better fit with a Bolgiano-Obukhov (BO) power law for stationary, stratified flows: $E(k) \sim \epsilon^{2/3} k^{-11/5}$, which we also include (Samuel & Verma, 2024).

At low Prandtl and Rayleigh numbers, momentum diffusivity is less dominant and buoyancy forces outpace viscous forces, therefore the solution can follow theoretical power laws for stationary turbu-

| Surface Variables (8) | Atmospheric Variables (5) | Forcing Variables (6) | Pressure Levels (13) |
|---|---|---|---|
| lwe of water evaporation (evap) | specific humidity (hus) | land sea mask (lsm) | 50, 100, 150, 200, |
| surface runoff (mrro) | air temperature (ta) | surface geopotential (sg) | 250, 300, 400, 500, |
| lwe of soil moisture content (mrso) | eastward wind (ua) | surface roughness length (z0) | 600, 700, 850, 925, |
| log surface pressure (pl) | northward wind (va) | TOA Incident Radiation (rsdt) | 1000 |
| 12h accumulated precipitation (pr_12h) | geopotential (zg) | sea ice cover (sic) | |
| 6h accumulated precipitation (pr_6h) | | sea surface temperature (sst) | |
| air temperature 2m (t2m) | | | |
| surface temperature (ts) | | | |

Table 13: Climate variables grouped into surface, atmospheric, and forcing variables. Additionally, the 13 pressure levels are used.

lence. At large Prandtl numbers and low Rayleigh numbers, momentum is dissipated more quickly, resulting in lower energy at most wavenumbers compared to stationary turbulence. At higher Rayleigh numbers, buoyancy forces created by the temperature gradient outpace viscous forces, resulting in energy injection with large-scale eddies that drive convective mixing. This added momentum is dissipated at large Prandtl numbers. In general, Rayleigh-Bénard Convection, while a turbulent system, does not universally match power laws derived for stationary turbulence due interactions between momentum dissipation, viscous forces, and buoyant forces.

While insightful, this doesn't necessarily guarantee that Rayleigh-Bénard Convection is not stationary, just that it does not mimic ideal, stationary cases. To test this, we calculate the autocorrelation of the time series of different fields (buoyancy, x-velocity, y-velocity, pressure) as well as the autocorrelation of a stationary Gaussian distribution, plotted in Figure 12. The autocorrelation for a given time lag $\tau$ is given as $C(\tau) = \langle u(t), u(t + \tau) \rangle$, or the average product of a random variable at two timesteps. Given large enough lag $\tau$, the autocorrelation should be constant over time for stationary distributions, as samples no longer depend on the absolute time $t$.

Depending on the Rayleigh and Prandtl number, Rayleigh-Bénard Convection can either be very far from stationary or approach a stationary distribution. Interestingly, at high Prandtl numbers, momentum diffusivity tends to damp velocity autocorrelations, while larger Rayleigh numbers tend to push systems away from stationary distributions by continuing to mix fluids through convection. While we cannot conclusively say much about Rayleigh-Bénard Convection, studying its statistical properties and its dependence on system parameters is still insightful. Indeed, this phenomena is still an active area of research with conflicting opinions (Lohse & Shishkina, 2024), which makes it an interesting and challenging case to build surrogate models for.

**PlaSim**  Simulations are solved using PlaSim (Lunkeit et al., 2021; Wikner et al., 2025), which assumes a set of governing equations for planetary climate based on the conservation of mass, momentum, and energy. Additionally, many variable-specific equations are used. A full description of the climate variables and the pressure levels used in the climate simulation dataset is given in Table 13. There are 8 surface variables and 5 atmospheric variables at 13 pressure levels, resulting in 73 prognostic variables. Additionally, 6 constant or yearly constant forcing variables are included as extra inputs.

## E  ADDITIONAL METHODS

### E.1  MODELS

**Generative Models**  DDPM, DDIM, TSM, and EDM models use some variant of the stochastic process:

$$x_t = \sqrt{\bar{\alpha}_t} x_0 + \sqrt{1 - \bar{\alpha}_t} z, \quad z \sim \mathcal{N}(0, \mathbf{I}) \tag{24}$$

which noises data $x_0$ over some time $t \in [1, T]$. While $x_t$ approaches $z$ as $t \to \infty$, this does not happen in finite time. This can cause inconsistencies when not using enough timesteps, and indeed, DDPM/DDIM models usually train with a fine discretization $T = [100, 1000]$ and require careful choice of noise schedule $\alpha_t$. To remedy this, flow matching uses the following stochastic process:

$$x_t = t x_0 + (1 - t) z, \quad z \sim \mathcal{N}(0, \mathbf{I}) \tag{25}$$

| | Training Objective | Sampling Procedure |
|---|---|---|
| **DDPM** | $\mathcal{L} = \|\epsilon - \epsilon_\theta(x_t, c, t)\|^2$ | $x_{t-1} = \frac{1}{\sqrt{\alpha_t}}\left(x_t - \frac{1-\alpha_t}{\sqrt{1-\bar{\alpha}_t}}\epsilon_\theta(x_t, c, t)\right) + \sigma_t z$ |
| **DDIM** | $\mathcal{L} = \|\epsilon - \epsilon_\theta(x_t, c, t)\|^2$ | $x_{t-1} = \sqrt{\alpha_{t-1}}\left(\frac{x_t - \sqrt{1-\alpha_t}\epsilon_\theta(x_t, c, t)}{\sqrt{\alpha_t}}\right)$ $+ \sqrt{1-\alpha_{t-1} - \sigma_t^2} \cdot \epsilon_\theta(x_t, c, t) + \sigma_t z$ |
| **EDM** | $\mathcal{L} = \|D_\theta(x + z, c, \sigma) - x\|^2$ | $x_{t+\Delta t} = x_t + \frac{\dot{\sigma}(t)}{\sigma(t)}\left(x - D_\theta(x, c, \sigma(t))\right)\Delta t$ (Euler) |
| **TSM** | $\mathcal{L} = \|\epsilon - \epsilon_\theta(x_t, c, t)\|^2$ | $x_0 = (x_t - \sqrt{1-\bar{\alpha}_t}\epsilon_\theta(x_t, c, t))/\sqrt{\bar{\alpha}_t}$ (1-step) |
| **FM** | $\mathcal{L} = \|(x - z) - v_\theta(x_t, c, t)\|^2$ | $x_{t+\Delta t} = x_t + v_\theta(x_t, c, t)\Delta t,\ x_0 \sim \mathcal{N}(0, I)$ (Euler) |
| **SI** | $\mathcal{L} = \left\|(\dot{I}(x_0, x_1, t) + \dot{\sigma}_t W_t) - b_\theta(x_t, c, t)\right\|^2$ | $x_{t+\Delta t} = x_t + b_\theta(x_t, c, t)\Delta t,\ x_0 \sim \rho_0$ (Euler) |

Table 14: Training and sampling for conditional diffusion, flow matching, and stochastic interpolant frameworks. Note that $x_t$ is defined differently for diffusion, flow matching, and stochastic interpolants. Samplers can also vary for EDM, TSM, FM, and SI.

where $t \in [0, 1]$. This process is exact at the endpoints $t = \{0, 1\}$, where $x_0$ is considered to be fully noised and $x_1$ is considered to be denoised. Under a linear choice of $\alpha(t), \beta(t)$, stochastic interpolants use the same stochastic process if the source distribution $x_0$ is chosen to be a Gaussian. However, admitting arbitrary source and target distributions allows stochastic interpolants to use the stochastic process:

$$x_t = (1-t)x_0 + tx_1 + (1-t)W_t, \quad W_t \stackrel{d}{=} \sqrt{t}z, \quad z \sim \mathcal{N}(0, \mathbf{I}) \tag{26}$$

where we make the appropriate choices for $\alpha(t), \beta(t), \sigma(t)$. In practice, to train generative models to learn and sample these stochastic processes, we use training objectives and sampling algorithms detailed in Table 14.

The Euler and Euler-Maruyama integrators are implemented by taking an initial sample $x_0$, either from Gaussian noise or the current state $u(t)$. A drift $dx_\theta(x_t, c, t)$ is calculated at time $t$ with the current sample $x_t$ and conditional information $c$, using either the EDM, FM or SI models. The Euler integrator updates the current state according to:

$$x_{t+1} = x_t + \Delta t * dx_\theta(x_t, c, t) \tag{27}$$

while the Euler-Maruyama integrator updates the current state with:

$$x_{t+1} = x_t + \Delta t * dx_\theta(x_t, c, t) + \sigma(t)\sqrt{\Delta t} * z \tag{28}$$

where $\Delta t$ is the step size, $\sigma(t)$ is some time-dependent scaling function for the noise, and $z \sim N(0, \mathbf{I})$. One note is that for stochastic interpolants, using an ODE-based sampler will result in a single prediction, as the sampling starts from the current state and no noise is used during integration.

We make a few implementation choices to stabilize the training and sampling of stochastic interpolants. We find that antithetic sampling (Albergo et al., 2023) helps to reduce the variance of the training loss and improves model performance. When $\dot{\sigma}(t)$ is singular, the variance of the loss can be infinite at the endpoints as $t \to 0$ or $t \to 1$. Antithetic sampling combines loss functions for the two stochastic processes $x_t^+ = I(x_0, x_1, t) + \sigma(t)W_t$ and $x_t^- = I(x_0, x_1, t) - \sigma(t)W_t$ to jointly learn the drift for both $x_t^+$ and $x_t^-$. This results in a finite variance as $t \to 0$ or $t \to 1$. Furthermore, following Chen et al. (2024), the first sampling step for the EM sampler is analytically computed to avoid potential numerical singularities in the probability flow SDE:

$$x_{\Delta t} = x_0 + \Delta t b_\theta(x_0, 0) + \sqrt{\Delta t}(1-t)z \tag{29}$$

**Model Architectures** For a given task (Kolmogorov Flow, RBC, Climate), the architecture is kept constant across all autoencoders and diffusion backbones. Parameter counts for Kolmogorov Flow are 21M for the FNO model, 20.5M for the autoencoder, and 57.9M for DiT backbones. Parameter

counts for RBC are 68.4M for the FNO model, 57.3M for the autoencoder, and 232M for DiT backbones. Parameter counts for PlaSim are 218M for the SFNO model, 89.5M for the autoencoder, and 313M for DiT backbones.

For autoencoders, we rely on convolution to process and downsample/upsample inputs. At each layer a standard Residual block processes inputs; at downsample/upsample layers PixelShuffle is used. Latent vectors are constrained either with a small KL regularization loss or a saturation function $z = \frac{z}{\sqrt{1+z^2/b^2}}$, where $b = 5$.

For the DiT backbone, the diffusion timestep $t$ is passed into the model with adaptive layer norm (AdaLN) after sinusoidal embedding. For datasets with extra scalar information (Rayleigh/Prandtl number, day of year/hour of day), it is added to the timestep embedding. For additional fields (PlaSim forcing variables), they are embedded and passed into the model with cross attention. Furthermore, every generative model is conditional since information is provided about the current state to sample a future state. To facilitate this, the noised state is concatenated along the channel dimension to the current state as input to the DiT.

## E.2 METRICS

**Variance-Scaled RMSE** Given a spatio-temporal data sample $\mathbf{u} \in \mathbb{R}^{n_t \times n_x \times n_y}$ and a model roll-out $\mathbf{u}_\theta \in \mathbb{R}^{n_t \times n_x \times n_y}$ the Variance-Scaled RMSE (VRMSE) is given by:

$$VRMSE(\mathbf{u}, \mathbf{u}_\theta) = \frac{1}{n_t} \sum_{t=1}^{n_t} \frac{||\mathbf{u}_\theta(t) - \mathbf{u}(t)||_2}{||\mathbf{u}(t) - \bar{\mathbf{u}}(t)||_2 + \epsilon}, \quad ||\mathbf{u}||_2 = \sqrt{\frac{1}{n_x n_y} \sum_{i=1}^{n_x} \sum_{j=1}^{n_y} |\mathbf{u}(i,j)|^2} \quad (30)$$

The term $\epsilon = 10^{-6}$ is added for numerical stability. This metric scales the error by the variance of the input sample, then averages over the prediction horizon $n_t$. The metric is more representative when nonnegative fields are present, as the more common Normalized RMSE (NRMSE) or Relative L2 Error tends to down-weight these channels. Additionally, predicting the mean field will result in $VRMSE(u, \bar{u}) \approx 1$, which is a useful interpretation.

**Spectral RMSE** Given a spatio-temporal data sample $\mathbf{u} \in \mathbb{R}^{n_t \times n_x \times n_y}$ and a model rollout $\mathbf{u}_\theta \in \mathbb{R}^{n_t \times n_x \times n_y}$ the DFT is used at each timestep to generate the power spectrum $p(t)$ and frequencies $k$. The power spectrum is partitioned based on its frequency into three evenly log-spaced bins. The SRMSE between the true and predicted spectra $(p, p_\theta)$ for each bin is then calculated as:

$$SRMSE(\mathbf{u}, \mathbf{u}_\theta) = \frac{1}{n_t} \sum_{t=1}^{n_t} \sqrt{1 - \left(\frac{p(t)}{p_\theta(t)}\right)^2} \quad (31)$$

For inputs with multiple channels, the SRMSE is calculated for each channel separately, then averaged.

**Latitude-Weighted RMSE** Latitude-weighted RMSE (lRMSE) is calculated at each lead time, for each variable and pressure level separately. For a forecasted variable $f_{tl} \in \mathbb{R}^{n_{lat} \times n_{lon}}$ and ground-truth $o_{tl} \in \mathbb{R}^{n_{lat} \times n_{lon}}$ at level $l$ and time $t$, lRMSE is given by Rasp et al. (2024):

$$lRMSE = \sqrt{\frac{1}{n_{lat} n_{lon}} \sum_{i=1}^{n_{lat}} \sum_{j=1}^{n_{lon}} w(i)(f_{tl}(i,j) - o_{tl}(i,j))^2}, \quad w(i) = \frac{\sin \theta_i^u - \sin \theta_i^l}{\frac{1}{I} \sum_{i=1}^{I} (\sin \theta_i^u - \sin \theta_i^l)} \quad (32)$$

For latitude weights $w(i)$, $i$ denotes the index of the discretized latitude, and $\theta_i^u$ and $\theta_i^l$ denote the upper and lower bounds of the cell at latitude $i$. Latitude weights are used to account for distortion at the poles in an equiangular grid, which would otherwise over-emphasize predicted values near the poles. Lastly, lRMSE is calculated for a given lead time (i.e., 10-day lRMSE) by initializing a forecast at each day in the validation set and averaging across all forecasts for that lead time.

**Climatological Biases** Climatological biases involve averaging over a long rollout to evaluate the consistency of a climate emulator. In particular, the 10-year bias is calculated for each variable and

pressure level by:

$$\text{Bias}_{\text{10-year}} = lRMSE(f_{avg}, o_{avg}), \quad f_{avg} = \frac{1}{n_t} \sum_{t=1}^{n_t} f_l(t) \tag{33}$$

where $n_t \approx 10 * 365 * 4$, since a forecast is made every 6 hours for each day in 10 years. Note that $f_l(t) \in \mathbb{R}^{n_{lat} \times n_{lon}}$, $f_{avg} \in \mathbb{R}^{n_{lat} \times n_{lon}}$, and lRMSE reduces over latitude and longitude.

**Continuous Ranked Probability Score**   Similar to lRMSE, CRPS is calculated at each lead time, and for each variable and pressure level separately, however with the addition of $M$ different ensemble members. CRPS makes use of the latitude-weighted mean absolute error (lMAE):

$$lMAE = \frac{1}{n_{lat} n_{lon}} \sum_{i=1}^{n_{lat}} \sum_{j=1}^{n_{lon}} w(i)|f_{tl}(i,j) - o_{tl}(i,j)| \tag{34}$$

CRPS for a given lead time $t$, variable, and level $l$ is given by:

$$CRPS_{tl} = \frac{1}{M} \sum_{m=1}^{M} lMAE(f_{tl}^{(m)}, o_{tl}) - \frac{1}{2M(M-1)} \sum_{m=1}^{M} \sum_{n=1}^{N} lMAE(f_{tl}^{(m)}, f_{tl}^{(n)}) \tag{35}$$

Intuitively, the first term penalizes deviations of the individual ensemble members from the ground truth, and the second term encourages spread between ensemble members. To calculate the CRPS for a given lead time, ensemble forecasts are initialized every three days for the validation year, and CRPS values at each lead time are averaged for across each forecast. An ensemble size of 32 is used.

**Spread-Skill Ratio**   The spread-skill ratio (SSR) is the ratio of the ensemble spread to the ensemble skill. The spread is calculated for a given time, level, and variable as the square root of the ensemble variance:

$$\text{Spread}_{tl} = \sqrt{\sum_{i=1}^{n_{lat}} \sum_{j=1}^{n_{lon}} w(i) var_m(f_{tl}^{(m)}(i,j))} \tag{36}$$

where $var_m$ calculates the variance over $M$ ensemble members. The ensemble skill is given by the RMSE of the ensemble mean:

$$\bar{f}_{tl} = \frac{1}{M} \sum_{m=1}^{M} f_{tl}^{(m)}, \quad \text{Skill}_{tl} = lRMSE(\bar{f}_{tl}, o_{tl}) \tag{37}$$

where $f_{tl}^{(m)} \in \mathbb{R}^{n_{lat} \times n_{lon}}$ is an individual forecast for a variable at lead time $t$ and level $l$. The SSR is given by:

$$SSR_{tl} = \frac{\text{Spread}_{tl}}{\text{Skill}_{tl}} \tag{38}$$

Calculating SSR for a given lead time is done similarly to CRPS, where SSR values for a lead time are averaged across all ensemble forecasts made for the validation year. SSR values less than 1 indicate an under-dispersive forecast, where the ensemble fails to capture the full range of possible outcomes, while values over 1 indicate an over-dispersive forecast, where the ensemble is overly uncertain.

### E.3   DISTANCE HEURISTICS

**Results for RBC**   For the SW and MMD metrics, Gaussian noise more closely resembles physical states after convective mixing (around $t = 20$), where there is a transition between laminar and turbulent regimes. This aligns with our understanding of turbulence as a multiscale and chaotic phenomena. However, trends for subsequent distributions are less clear. At the onset of turbulence and at certain Rayleigh and Prandtl numbers, the system can become stationary. In this case, classifiers would not be able to differentiate subsequent distributions, as reflected by the low C2ST accuracy. However, this would suggest that the SWD/MMD metrics report artificially large distances, which is a known problem at low sample sizes (Bischoff et al. (2024), Figure 8). Indeed, at a fixed $Ra$ and $Pr$ number, we only have access to 40 samples at each timestep to evaluate distances between $u(t)$ and

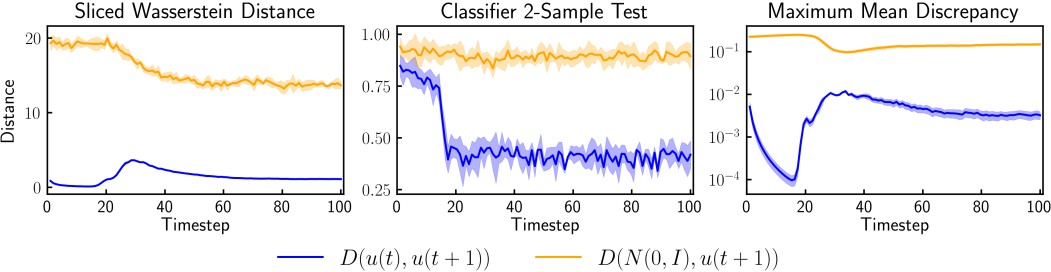

Figure 13: Distance heuristics for the Rayleigh-Bénard dataset at $Ra = 1e10$ and $Pr = 0.1$. Distances between samples drawn from successive timesteps $D(u(t), u(t+1))$ and between Gaussian noise and future timesteps $D(N(0, I), u(t+1))$ are plotted for the buoyancy field. Each metric is averaged over a 5-fold cross validation, with the standard deviation shaded.

$u(t+1)$. However, if the distribution is still changing in time, this would be captured by SWD/MMD metrics but not C2ST. Bischoff et al. (2024) report that classifiers struggle to learn at small dataset sizes, which can underestimate C2ST accuracy. In this case, increasing the classifier size did not seem to increase C2ST accuracy as well. Based on the complexity of the underlying distribution as well as the low sample count, it is difficult to make claims about distances between subsequent distributions. However, the primary takeaway remains unchanged that distances between a Gaussian and physical states are larger when measured by all metrics than distances between subsequent states.

**Calculation** Considering each timestep of a PDE/climate trajectory as a distribution is not a common perspective, although it is often implicitly assumed when applying generative models to these emulation tasks. As such, we seek to build some intuition on this perspective through a set of visualizations, in Figures 14 and 15.

We use t-SNE to visualize samples from the Kolmogorov Flow or Rayleigh-Bénard Convection datasets. This is purely for visualization and intuition, no claims about distances or distributions can be made based on the plots. Interestingly, t-SNE can portray initial distributions based on what we expect, since we know the distributions that are used to sample randomized initial conditions. Over time, the initial distribution may be transported over time based on the PDE, which we visualize both for all samples and a single trajectory. At each timestep, we don't know this true distribution or if it even exists, however, we have access to some of its samples. Therefore, we look to quantify a distance between two subsequent empirical distributions. Furthermore, since we can sample a Gaussian, we also look to quantify a distance between a Gaussian and an empirical distribution.

Heuristics for calculating distances between distributions where we only have access to samples exist (Bischoff et al., 2024), although their quality may be impacted by many factors. One factor is the dimensionality of samples drawn from the considered distributions. High-dimensional distributions are more challenging to work with and heuristics are less accurate, therefore we first use a dimensionality reduction that preserves distances, based on the Johnson-Lindenstrauss Lemma.

Consider a set of $n$ samples $\{x^1, x^2, \ldots, x^n\}$, $x^i \in \mathbb{R}^d$, where $d$ can be very large. In our case, this can be a flattened sample $u(t)$. We define a reduced dimension $m < d$ and a random projection matrix $P \in \mathbb{R}^{m \times d}$, where each entry is sampled from a normal Gaussian $P_{ij} \sim \mathcal{N}(0, 1)$. $P$ is additionally scaled to obtain $\hat{P} = \frac{1}{\sqrt{m}} P$. Consider a Euclidean distance on vectors $||x||_2 = \sqrt{\sum_{i=1}^{d} x_i^2}$. Given some error $\epsilon$ if $m = O(\frac{\log n}{\epsilon^2})$, then:

$$(1 - \epsilon)||x^i - x^j||_2^2 \leq ||\hat{P}x^i - \hat{P}x^j||_2^2 \leq (1 + \epsilon)||x^i - x^j||_2^2 \tag{39}$$

Fortunately, the choice of $m$ does not depend on the original dimension $d$, which is beneficial if $d$ is large. We can therefore leverage this to preserve pairwise Euclidean distances between $x^i, x^j$ while reducing the dimensionality of the samples. This is assuming we have enough samples and choose modest error bound $\epsilon$, which we set to $\epsilon = 0.2$. After projecting each sample to a lower dimension, we use implementations from Bischoff et al. (2024) to calculate the Sliced Wasserstein

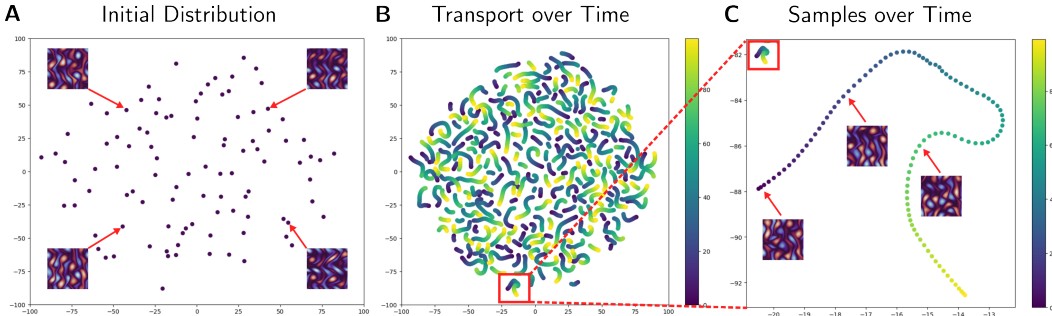

Figure 14: t-SNE visualizations for samples from Kolmogorov Flow. *Left, A:* We plot initial conditions $u(t = 0)$ from the entire dataset at after dimensionality reduction using t-SNE. As expected, the initial distribution is roughly uniform as Fourier coefficients used for initial conditions are uniformly sampled. *Middle, B:* Samples from the entire dataset are plotted, where each sample can be from a different initialization or timestep. Samples are colored by their timestep, where lighter colors are later timesteps. *Right, C:* A single trajectory is enlarged and visualized. There is some path that transports a single initial condition through time. As a whole, there may be a distribution at each timestep that is transported through time.

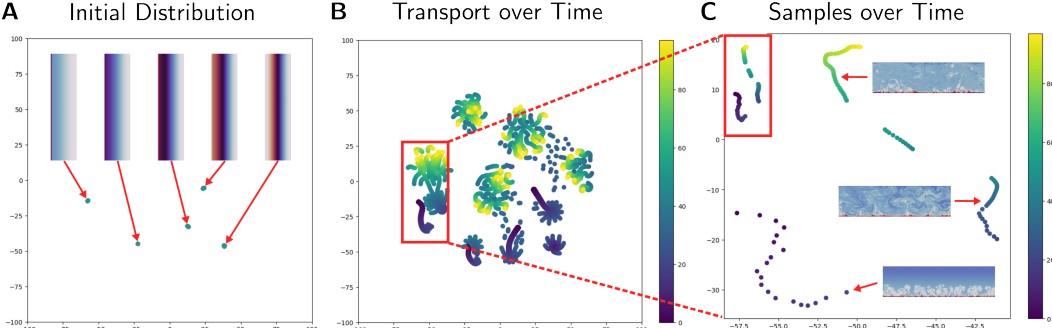

Figure 15: t-SNE visualizations for samples from Rayleigh-Bénard Convection. *Left, A:* We plot initial conditions $u(t = 0)$ from the entire dataset at after dimensionality reduction using t-SNE. As expected, the initial distribution is roughly categorical, as initial condition coefficients $\delta b_0$ are sampled from $\{0.2, 0.4, 0.6, 0.8, 1.0\}$. *Middle, B:* Samples from the entire dataset are plotted, where each sample can be from a different initialization or timestep. Samples are colored by their timestep, where lighter colors are later timesteps. Although samples are instantiated at similar initial conditions, chaotic mixing causes different instantiations to diverge. *Right, C:* A single trajectory is enlarged and visualized. There is some path that transports a single initial condition through time, although perhaps not easy to visualize. As a whole, there may be a distribution at each timestep that is transported through time.

Distance, Classifier 2-Sample Test, and Maximum Mean Discrepancy between distributions, where each distribution is represented by $n$ samples at a given timestep. Additionally, we perform 5-fold cross validation by taking $80\%$ of the total samples at each timestep as the empirical distribution.

