# OpenReview forum: "Reframing Generative Models for Physical Systems using Stochastic Interpolants"
_ICLR.cc/2026/Conference — Submitted to ICLR 2026_

### Official Review · Reviewer_Todc · 2025-10-29

**Soundness:** 2
**Presentation:** 3
**Contribution:** 1
**Rating:** 2
**Confidence:** 4

**Summary:**

This paper benchmarks generative surrogates for physical systems and argues for stochastic interpolants(SI) that learn a process directly between current and next states, enabling fewer sampling steps and improved accuracy/calibration for autoregressive forecasting. The authors conduct a comprehensive benchmark of SI against other leading generative models (DDPM, DDIM, EDM, Flow Matching) and deterministic baselines (FNO/SFNO). The evaluation is performed on three diverse and challenging physical datasets: Kolmogorov Flow (PDE), Rayleigh-Bénard Convection (chaotic PDE), and PlaSim (global climate model). SI is positioned as a competitive baseline for physical emulation, with tunable ODE/SDE sampling to balance pointwise accuracy v.s. spectral fidelity.

**Strengths:**

- Originality. This paper performs autoregressive physical emulation using stochastic interpolants (SI) that evolve samples directly from current to future states, unifying ODE/SDE sampling and avoiding Gaussian-noise conditioning when source/target are close. This is well-motivated for PDE/weather.
- Quality. This paper uses matched backbones/latents across methods, reports NFEs, and evaluates on three representative datasets with pointwise, spectral, and probabilistic metrics; includes ablations (pixel vs latent, compression ratio, deterministic vs probabilistic).
- Clarity. This paper provides a concrete SI definition with boundary conditions and a clear training loss, plus explicit ODE/SDE sampling equations. The metrics are defined in the paper/appendix for reproduce evaluation.

**Weaknesses:**

## Contradiction and misinterpretation in the paper's motivational evidence
Theoretically, after the initial transition phase/ turbulent mixing (around t=20 in the Rayleigh-Bernard convection case), the system is expected to enter a statistically stationary regime. In this state, while the instantaneous $u(t)$ is chaotic, the overall distribution $p(u(t))$ should remain basically unchanged.
- The paper's own SWD and MMD scores in Figure 2 are in direct contradiction with physical principles and C2ST score. Base on SWD and MMD curves, the paper states, *"subsequent states seem to be farther during and after this transition".* This suggests the distributions are "far" apart, which is unphysical and should not have happened.
- At the statistically stationary stage, the C2ST score stays at ~50%. The paper sees the 50% score and provides this explanation: *"Interestingly, classifiers struggle to take advantage of this; subsequent timesteps have ~50% classification accuracy in turbulent regimes, as consistent changes between states become harder to learn"*.  As discussed, the observation that the classifier "struggle to take advantage of" is unphysical and should not have happened. Also, A 50% score does not mean that changes between turbulence states are hard to learn. It means the classifier has failed because the two distributions have become indistinguishable. This is not a failure of the metric, it is a strong evidence for the paper's hypothesis. It shows that in the statistically steady turbulent regime, the distribution at time t is so similar to the distribution at t+1 that a powerful classifier cannot find any signal to tell them apart.

**The authors  fail to discuss this glaring contradiction, and also misinterpret the one metric (C2ST) that actually supports their physical intuition.**

## Insufficient contribution beyond hyperparameter tuning and the selective application of known techniques.
The paper's core contribution reduces to: "We applied stochastic interpolants (an existing method) to physical emulation (where similar generative models already exist) and found they work competitively with appropriate hyperparameter tuning."

-  **What is already known** The complete mathematical framework, training objectives, sampling procedures, ability to adjust during sampling without retraining, data-dependent couplings, are already fully established. (e.g. Albergo & Vanden-Eijnden 2023, Albergo et al. 2023, Chen et al. 2024, Albergo et al. 2024). The trade-offs between deterministic/stochastic sampling and the success of interpolants/bridges in related domains are already known. The observation that transporting related distributions is easier than transporting from Gaussian noise is the on of the foundational motivations of the the entire bridge/interpolant literature (e.g. Zhou et al. 2023, Su et al. 2023, Albergo et al. 2024, Zheng et al. 2025, Bortoli et al. 2023, Tong et al. 2024). This is not a new insight. The applications of SI to physics (especially, forecasting and fluids) are not new.

- **What the paper contributes** Empirical results showing SI is competitive (sometimes better, often comparable, but **without error bar**) on three physical systems; Ablations on hyperparameters; Distance heuristics visualizations which demonstrate a known issue.

- **The  "Win" for SI is attributable to tuning** The paper only demonstrates that a hyper-tuned SI can beat default-tuned baselines.
    - The paper's own ablation study (Table 9) is the key piece of evidence. For example, it shows that SI's performance on Kolmogorov flow is critically dependent on the noise coefficient $\sigma$. At $\sigma=3$, its NRMSE is the worst of any generative model in Table 1. At $\sigma=1$, its NRMSE is the best score reported in Table 1. While a valid empirical study would apply this same level of tuning to all competitors. The paper does not. It explicitly states it uses the default, non-optimized schedules for its baselines, such as "the noise schedule for EDM is reproduced from Karras et al. (2022)" and "For FM models, we use the rectified flow schedule". **This unequal tuning invalidates the paper as a fair empirical study**.

- **The paper's "Solutions" are selective applications of known tricks**
    - The paper's solution for the accuracy-vs-statics trade-off is to adjust the sampling (i.e. switch between an ODE and SDE sampler). But this flexibility is a known, fundamental property of all continuous-time generative models, including the EDM baseline. A true empirical study would have applied this "trick" to all capable models.

To conclude, the paper suffers from a positioning crisis, as it is not a new method, not a novel application, and fails as a fair benchmark/empirical study.

## Statistical robustness.
The paper's conclusions are not statistically robust. Kolmorogov flow experiment is under-sampled. The results are reported over 3 seeds, which is insufficient to robustly capture the variance inherent in generative models. For Rayleigh-Benard convection and PlaSim, only one run is performed. Main tables except table 1 lack error, probabilistic metrics are averaged but no bands are shown. Therefore, the conclusions about method ranking are not statistically supported. This could be a material limitation for more chaotic/turbulent tasks.

**Questions:**

Some minor concerns:
- Scope narrower than the abstract's claim of "diverse physical domains." The work benchmarks three close related fluid/climate settings - Kolmogorov flow, Rayleigh-Benard convection, and the challenging PlaSim climate emulation. That's diverse within fluids, but not across broader physical domains. For the simplest dataset, Kolmogorov Flow, the authors report error and standard deviation over only three seeds, and then reduce that to a single run for their more complex, chaotic datasets. For Kolmogorov flow, this is insufficient to capture variances
- Spectral RMSE definition is inconsistent between main tex and appendix. The definition in Appendix D.2 misses the square.
- The paper cites Lancelin et al. (in prep) and Wikner et al. (in prep). There are two issues. The first one is, these works are not publicly available and hence unverifiable. The second one is, **how would the authors of this submission be among the few aware of these unavailable "in prep" works and their title/contents?**

---

> ### Author Response · Authors · 2025-11-18
> **Response to Reviewer Todc**
>
> Thank you for the thoughtful comments and detailed review. Your feedback has prompted us to reconsider many positions taken throughout the work and introduce some new experiments. We hope that these efforts have improved the quality of this work, which we have uploaded as a revised manuscript.
>
> > Contradiction and misinterpretation in the paper's motivational evidence
>
> Thank you for the comment, and your concern has prompted us to take a closer look at the statistical properties of Rayleigh-Benard Convection. Turbulence can indeed be a stationary phenomena, but there are many types of turbulence (forced, decaying, isotropic, wall-bounded, buoyancy-driven, etc.) where not all of them are stationary or even well-understood. However, a well-understood case is stationary, homogeneous and isotropic turbulence, whose energy spectrum can be fit with a power law, such as the familiar Kolmogorov-Obukhov (KO) $\epsilon^{2/3}k^{-5/3}$ law. Additionally some evidence suggests RBC is better fit with the Bolgiano-Obukhov (BO) $e^{2/3}k^{-11/5}$ law for stationary, stratified flows [1]. As such, we look to compare different regimes of RBC flow to these theoretical correlations, in Figure 11 of the updated manuscript.
>
> At low Prandtl and Rayleigh numbers, momentum diffusivity is less dominant and buoyancy forces outpace viscous forces, therefore the solution follows theoretical power laws for isotropic, forced turbulence better. At large Prandtl numbers and low Rayleigh numbers, momentum is dissipated more quickly, resulting in lower energy at most wavenumbers compared to stationary turbulence. At higher Rayleigh numbers, buoyancy forces created by the temperature gradient outpace viscous forces, resulting in energy injection with large-scale eddies that drive convective mixing. This added momentum is dissipated at large Prandtl numbers. In general, Rayleigh-Benard Convection, while a turbulent system, does not universally match power laws derived for stationary turbulence due to interactions between momentum dissipation, viscous forces, and buoyant forces.
>
> While insightful, this doesn’t necessarily guarantee that RBC is not stationary, just that it does not mimic ideal, stationary cases. To test this, we calculate the autocorrelation of the time series of different fields (buoyancy, x-velocity, y-velocity, pressure) as well as the autocorrelation of a stationary Gaussian distribution in Figure 12 of the updated manuscript. For large enough lag $\tau$, stationary distributions exhibit constant autocorrelations, which we observe in some cases. However, depending on the Rayleigh and Prandtl number, the RBC system can be very far from stationary. Interestingly, at high Prandtl numbers, momentum diffusivity tends to damp velocity autocorrelations, while larger Rayleigh numbers tend to push systems away from stationary distributions by continuing to mix fluids through convection.
>
> In reality, the statistical properties of RBC are more nuanced than a stationary distribution, and to exacerbate this, we used buoyancy fields sampled from a set of simulations with Rayleigh=1e10 and Prandtl=0.1. However, we still believe your interpretation has merit. In stationary cases, C2ST would be correctly reporting close distances while MMD/SWD would be overestimating distances between subsequent distributions. This is a known failure mode of MMD/SWD in small sample-size regimes ([2], Figure 8). Indeed, we only have access to 40 samples at a given Ra/Pr number and at each timestep. However, when the dynamics are not stationary, MMD/SWD would be correctly capturing this, while C2ST would be underestimating the distances. Interpreting distance from a classifier can have many confounding factors, such as the size of the classifier and dataset size. At small sample sizes, it is also known that C2ST accuracy is lower ([2], Figure 5), and the classifier we trained is also small (2-layer MLP). Despite these nuances, the main takeaway from the motivating experiments remains unchanged, that distances between Gaussian and flow fields are farther than distances between subsequent flow fields.
>
> However, we also recognize the fragility of sample-based distance heuristics, and the complexity of the RBC system that results in this inconclusive result. Therefore, we have decided to move this discussion of RBC to the Appendix and instead consider running these metrics for Kolmogorov Flow, which we have 1024 samples at each timestep and is a well-studied, stationary flow [3]. Furthermore, the Re number is constant for all Kolmogorov Flow samples. In this setup, we both know more about the underlying distribution and the metrics have fewer failure modes, which we hope will be a more rigorous motivating example. All told, we have changed Figure 2, modified Section 4, and incorporated this feedback in Appendix C and D.3, which we hope will clarify things.

---

> ### Author Response · Authors · 2025-11-18
> **Response to Reviewer Todc (Continued)**
>
> > The applications of SI to physics (especially, forecasting and fluids) are not new.
>
> We agree with this point and have been transparent about our contribution by citing all of the mentioned works in our related works. We agree that the application of SI to climate forecasting/fluids is not new, however, through empirical studies we are trying to understand more about it. To this end, we hope to be more clear about our contribution by renaming the work to “Benchmarking Stochastic Interpolants for Physical Emulation”. We hope that our experiments can help give insight to when stochastic interpolants work well and why these capabilities are observed.
>
> > This unequal tuning invalidates the paper as a fair empirical study.
>
> Thank you for bringing this concern to our attention. We apologize if we have given the impression that the baselines were not tuned, as during our experimentation, we have made substantial effort to both reproduce best-practices in the field for generative models for physics as well as independently tune baseline parameters. The use of the rectified flow/optimal transport schedule is in line with the state of the art for PDE modeling [4, 5, 6]. Even in the image generation, the rectified flow schedule is shown to outperform other schedules [7]. There are also no hyperparameters to tune in the rectified flow schedule (beyond the NFEs).
>
> We also have results for hyperparameter tuning for Kolmogorov Flow, where we experimented with different noise schedules for DDPM, DDIM, TSM and EDM. We excluded these in the main paper due to the already lengthy ablation studies section but are happy to reintroduce them, as well as complement these initial runs with additional hyperparameter tuning for our baselines. These can be found in Tables 10 and 11 in the revised work. In general, we find that best practices from prior work transfer reasonably well. Prior works [8,9] suggest that linear noise schedules work well for DDPM-based models, which we independently verify and tune a scaling factor for. Additionally, the EDM noise schedule works well, likely since it is highly optimized by the original authors and we normalize our data to a similar scale.
>
> Furthermore, we only experimented with hyperparameters in the Kolmogorov Flow experiment due to computational constraints. The results of Rayleigh Benard and all Climate results are run with a single set of hyperparameters, as training a model on these datasets requires around 6 A100-days. We hope that this eases concerns of fairness.
>
> > The paper's own ablation study (Table 9) is the key piece of evidence. For example, it shows that SI's performance on Kolmogorov flow is critically dependent on the noise coefficient.
>
> We are unsure if Table 9 is a detriment, as any generative model is critically dependent on its noise schedule. If the amount of noise in any diffusion model were tripled it would certainly be much worse; this doesn’t mean that the generative model is bad.
>
> Furthermore, you may notice that the optimal coefficient $\sigma =1$. This is in line with what we expect, that the scale of the $\alpha(t), \beta(t), \sigma(t)$ terms are the same in the interpolant, and is the default setup in prior works [10]. We reported Table 9 not as a hyperparameter sweep to ‘optimize’ over SI parameters, but as a way for readers to understand the role of Gaussian noise in an interpolant. Indeed, we used the default setup (no scaling factor or $\sigma=1$) in all our experiments and then decided to test a hypothesis about why we need Gaussian noise to mix the source and target distributions by introducing the scaling term $\sigma$ to modulate its effect.
>
> > The paper only demonstrates that a hyper-tuned SI can beat default-tuned baselines.
>
> Beyond applying our best efforts to maintain a fair comparison and tune our baselines, we believe that there are useful contributions of the work. There are other reasons for why a stochastic interpolant would outperform a baseline beyond just tuning, such as to the proximity of the modeled distributions. Furthermore, most practitioners are not aware of these potential benefits; therefore, we try our best to facilitate a balanced discussion about this through a range of experiments and baselines.

---

> > ### Author Response · Authors · 2025-11-18
> > **Response to Reviewer Todc (Continued)**
> >
> > > The paper's "Solutions" are selective applications of known tricks
> >
> > Thank you for raising this concern. We apologize if the language was misleading, since we are not advocating that SI models are better in this sense, as you mention that all generative models can have some level of accuracy vs. statistical tradeoff. In fact, we do apply this ‘trick’ to diffusion models, where DDPM uses an ancestral SDE-based sampler, DDIM uses an ODE-based sampler, and TSM uses a truncated sampler for the same trained model. We provide additional experiments of sampling the EDM model with an SDE-based sampler as well on Kolmogorov Flow and RBC systems (Table 2 and Table 11). We forgo applying this ‘trick’ to flow matching, since the point of rectified flow is to “is to learn the ODE to follow the straight paths connecting the points drawn from $\pi_0$ and $\pi_1$ as much as possible” [11], and introducing an SDE formulation defeats this purpose.
> >
> > Our intention is not to selectively apply this to SI models, but to point out that a sampling tradeoff is important when training generative models for physical systems. To this end, we will make this clear that it is a property of all generative models and have revised the abstract and claims in the paper.
> >
> > An additional note is that many of our results are not dependent on this ‘trick’. Using an ODE-based sampler in 10-day Climate forecasts and Kolmogorov Flow is faster and generally more accurate across all models, which we hypothesize is due to the unimodal conditional distributions present in these physical systems. Applying an SDE-based sampler to the next best ODE-based models of flow matching or EDM would only degrade performance.
> >
> > > To conclude, the paper suffers from a positioning crisis, as it is not a new method, not a novel application, and fails as a fair benchmark/empirical study.
> >
> > We have attempted to be more clear about our experiments and resolve concerns around fairness, however, if there are lingering concerns please let us know. The method is not new, as you mention, and therefore we have clarified our contribution to be a work about benchmarking by revising the title and abstract.
> >
> > > Statistical robustness.
> >
> > Thank you for raising this concern. We believe that the paper’s evaluations are in line with the standards of the field. As datasets have become larger and more complex, it has become less feasible to train multiple seeds for each model. There are a wealth of prior works that, despite only running one seed per model, have valuable insights [12, 13]. Even works that are tailored as benchmark studies can only run one, or at most three, different random seeds [4,9]. Given a finite compute budget, we choose to focus on evaluating more baselines and more challenging PDE/climate tasks, rather than more seeds. A single RBC or climate emulator takes 6 A100-days to train, and across 5 different models that is 30 A100 days. Adding multiple seeds would increase that to 60-100 A100 days. We understand the intention behind wanting more random seeds, but even if we wanted to run additional random seeds for this revision, we do not have the resources to.
> >
> > > Scope narrower than the abstract's claim of "diverse physical domains."
> >
> > Thank you for raising this point. We have updated the phrasing to “we benchmark generative models across fluid and climate systems”.
> >
> > > Spectral RMSE definition is inconsistent between main tex and appendix.
> >
> > Thank you for pointing this out, we have remedied this error.
> >
> > > The paper cites Lancelin et al. (in prep) and Wikner et al. (in prep).
> >
> > Thank you for raising this concern. One of the papers has been made available (Lancelin et al.). Wikner et al. has been removed from the paper.
> >
> > We apologize for the lengthy response and thank you for your time. We hope that we have addressed some of your concerns and are happy to clarify or continue discussions as necessary.
> >
> > References
> > 1. https://journals.aps.org/prfluids/abstract/10.1103/PhysRevFluids.9.023502
> > 2. https://arxiv.org/abs/2403.12636
> > 3. https://journals.aps.org/pre/abstract/10.1103/PhysRevE.89.023004#s2
> > 4. https://arxiv.org/abs/2507.02608
> > 5. https://arxiv.org/abs/2503.22600
> > 6. https://arxiv.org/abs/2506.03111
> > 7. https://arxiv.org/abs/2403.03206
> > 8. https://openreview.net/forum?id=0FbzC7B9xI
> > 9. https://arxiv.org/abs/2309.01745v3
> > 10. https://arxiv.org/abs/2303.08797
> > 11. https://arxiv.org/abs/2209.03003
> > 12. https://arxiv.org/abs/2403.03542
> > 13. https://arxiv.org/abs/2402.02366

---

> > > ### Comment · Reviewer_Todc · 2025-11-27
> > >
> > > The authors have provided a comprehensive and high-quality rebuttal that addressed my primary concerns regarding the validity of the motivational evidence and the fairness of the experimental comparison. I have revised my score.
> > >
> > > While the paper is now experimentally complete, its core contribution remains the application of existing methods  to established tasks. The work is a valuable empirical study for the AI for Science community, offering insights into model selection and distribution transport. However, the lack of fundamental methodological innovation prevents a higher score. Additionally, while the computational constraints are understandable, the absence of error bars for the complex climate/RBC datasets remains a minor statistical limitation.

---

### Official Review · Reviewer_KFnG · 2025-11-05

**Soundness:** 2
**Presentation:** 3
**Contribution:** 2
**Rating:** 4
**Confidence:** 4

**Summary:**

This paper describes its contribution as reframing the generative modeling of physical systems through the modeling approach of Stochastic interpolants.  Specifically the authors focus on the task of conditionally modeling the next time step solution u(t+1) of systems described by PDEs, given the current timestep u(t). The authors compare various generative modeling methods, including recent variants of stochastic interpolants, for a number of PDEs describing Kolmogorov flows, RB convection, and climate/weather. The authors emphasize the utility of SDE generative models whose initial condition starts at the current solution of the PDE, rather than from a Gaussian.

**Strengths:**

Thorough benchmarking of a wide array of generative modeling methods with a lot of reproducibility details mentioned, on several challenging PDE simulation problems.

**Weaknesses:**

I believe the paper exhibits great work on benchmarking existing generative modeling methods on a number of challenging tasks. However, I believe the contributions of the work could use some reframing. After reading, I’m not left with the impression that I have seen a reframing, or a new method proposed. In particular, while section 2 is a review of stochastic interpolants, section 3 “Methods”  mostly describes datasets, baselines, autoencoders, architectural choices, and metrics.

What I don’t see is much of a discussion of “reframing” in the paper in the form of something like
- it’s a long standing trend that people do XYZ, and we show evidence here that we should instead look at the problem as XYZ’
- or solve it in a new way with method XYZ’ that we propose
- or solve it with method XYZ’ that has been overlooked as a solution.

At a few points during the paper, an approach is emphasized: conditionally generating the u(t+1) from an interpolant with initial condition u(t), but I see that this has been explored in a few papers including the work by Chen et al (https://arxiv.org/abs/2403.13724) that the authors cite. I would also understand more if there was a framing of this work was that the observations in Chen et al (and related papers) are shown in this paper to extend to new types of data.

From the reviewing perspective, I would really benefit from one of the following:

- clarification on the nature of reframing or contributions that I may have missed
- an attempt to narrate the paper more closely with what it accomplishes, which is a super solid benchmarking of generative modeling methods on hard PDE tasks. But if so, please relate this to either works that have also done this (for example, https://arxiv.org/abs/2410.16415).
- or perhaps a clarification that the work being done is showing that existing methods extend to new instances of data.

If the authors feel that these two options are a mischaracterization, or inappropriately miss another aspects of the work, I’m open to hearing about it.

Regardless, I also proceed with a few more specific technical comments in "Questions".

In summary, I feel rather positively about the work done, but I feel that it is not what the paper describes as the work being done. I am open to discussion about it with the authors and other reviewers.

**Questions:**

**Question #1**

I have doubts that the SDE you present in (4):

$$
dX = b dt + \frac{\dot \gamma(t)}{\sqrt{t}}dW_t
$$

has the same time-varying marginals as the stochastic interpolant and the ODE.

I'll proceed by direct comparison to that in (Albergo et al) and in (Chen et al). Let us assume the setup with $\alpha, \beta, \gamma$ from (https://arxiv.org/abs/2303.08797). Let $b()$ denote the ODE velocity/drift that you (implicitly) define through the loss function in (3) as

$$
b(t, x) = E[\dot \alpha_t x_0 + \dot \beta_t x_1 + \dot \gamma_t z | x_t = x]
$$

Then it is true that the ODE you present, $dX = b dt$, starting at a sample of $x_0 \sim \rho_0$, passes through the same time marginals $\rho_t$ that the interpolant does.  However, according to (Albergo et al), the SDE with solutions that pass through these same marginals is, for some $\delta(t) > 0$,

$$
dX = f(t, x) dt + \sqrt{2 \delta(t)} dW_t, \quad f(t, x) = b(t, x) + \delta(t) s(t, x)
$$

Here $s(t, x) = \nabla \log \rho_t(x)$. The score can be shown to be $s(t,x)=-\frac{1}{\gamma(t)}E[z | x_t]$ so altogether:

$$
f(t, x) = E[\dot \alpha_t x_0 + \dot \beta_t x_1 + (\dot \gamma_t - \frac{\delta(t)}{\gamma(t)} )z | x_t = x]
$$

It doesn't immediately seem obvious to me that this generic $f()$ recipe equals the $b()$ mentioned in your (4) for the SDE.

Next, in (Chen et al 2024), which you mention just after your equation (4), they do make some special choices that yield an SDE that appears different to this recipe, but, can be shown to be an instance of that in (Albergo et al). Let us next review this choice., assuming their SDE has been proven to pass through the right marginals. (Chen et al) describe the interpolant directly via the Brownian motion rather than the gaussian $z$. They define it as

$$
x_t = \alpha(t) x_0 + \beta(t) x_1 + \sigma(t) W_t
$$

where $W_t$ is the time $t$ solution of a brownian motion/Wiener process which has the same distribution as $\sqrt{t}z$ for standard gaussian $z$. Therefore this interpolant has the same distribution as
$$
x_t = \alpha(t) x_0 + \beta(t) x_1 + \sigma(t) \sqrt{t} z
$$
for gaussian $z$, so we can see that it corresponds to the choice of your $\gamma(t) = \sigma(t) \sqrt{t}$ for some chosen $\sigma(t)$. We also note that this same $\sigma(t)$, which appears in the interpolant, also appears in (Chen et al)'s generative SDE, directly as the diffusion coefficient.

Next, In your section 3.2 (which is somewhat after your equation 4 where the SDE is given a very particular form), you set $\alpha(t)=1-t, \beta(t)=t$, and $\gamma(t)=(1-t)\sqrt{t}$. By your choice of $\gamma$ and by the previous reasoning step that $\gamma$ relates to $\sigma$ via $\gamma(t) = \sigma(t) \sqrt{t}$, we can conclude that the $\sigma(t)$ that is implicitly in play in your work is $\sigma(t)=1-t$, like in (Chen et al). However, (Chen et al) show that the appropriate SDE has diffusion coefficient exactly equal to $\sigma(t)$ i.e., $dX = b dt + \sigma(t) dW_t$. But the diffusion coefficient you choose in (4) is $\frac{\dot \gamma(t)}{\sqrt{t}}$. So we must next compare these choices. Taking $\gamma = (1-t)\sqrt{t}$ (as you choose in section 3) and computing your $\frac{\dot \gamma(t)}{\sqrt{t}}$ seems to give a diffusion coeffient of $\frac{1-3t}{2t}$, while (Chen et al) just had the diffusion coefficient $\sigma(t) = 1-t$.

Therefore, with the background of Albergo et al (https://arxiv.org/abs/2303.08797) and Chen et al (https://arxiv.org/abs/2403.13724), I don’t quite have the tools to see why the SDE in your (4) has solutions following the marginals of the interpolant and the ODE (thereby yielding the integration of this SDE as a method that samples from the generative model being trained)

Regardless of whether my reasoning about the mismatch holds up (sorry if any gaps, please let me know), could you please provide some sort of explanation (preferably proof) of how the SDE samples the desired distribution? If the specific form only holds under choices of $\alpha,\beta,\gamma$ (and maybe an implicit $\delta(t)$) used later on in the methods/experiments, could you please make sure to state these choices right next to (4) and clarify whether it only holds under those choses?

**Question #2 (relatively minor stuff)**

Some more confusion on ODE and SDE for stochastic interpolant:

- table 11 (appendix) reportedly describes the sampling procedures for each model, and only shows the Euler discretization of the ODE generative model defined by the interpolant. However, in the main text, the ODE (sampled with Euler) and SDE (sampled with Euler-Maruyama) are shown in the results tables. Could you please update table 11 to include the stochastic interpolant as well, or explain why it’s not listed.

- for the ODE: if the previous time step u(t) of the PDE solution does not determine the next step u(t+1) (either due to stochastic forcing or unobserved variables), then there is a distribution of u(t+1) given u(t). This is seemingly your motivation for using generative models. However, the ODE does need stochasticity in the usual random gaussian initial condition in order to model a whole distribution. How can the ODE starting at u(t) model a distribution?


**small comment**

Since velocity-trained stochastic interpolant with no gamma function and gaussian base distribution is equal to flow matching, especially when both are sampled with the ODE, could you help the reader by the reminding them that the difference is the $\gamma$, the gaussian base, and the choice of sampling algorithm, and especially provide clarity when flow matching and the interpolant Euler ODE show up in the same result tables?

---

> ### Author Response · Authors · 2025-11-18
> **Response to Reviewer KFnG**
>
> Thank you for your detailed review of our work! We have revised our manuscript with the suggested changes, which we have uploaded, but in summary:
>
> >  I feel rather positively about the work done, but I feel that it is not what the paper describes as the work being done. I am open to discussion about it with the authors and other reviewers.
>
> Thank you for the detailed comment. We would agree with you, and have decided to change the title of the work to “Benchmarking Stochastic Interpolants for Physical Emulation”. The intention behind the original “reframing” is that most practitioners in the field continue to use generative models based on Gaussian denoising, mostly since they are more popular and better known than interpolant-based approaches. However, we recognize that this is potentially misleading and have worked to reframe the "Reframing" (haha) to fix this. In particular:
>
> - The title is changed (Updates to the title can only be made after the reviewing deadline, but it's updated in the new manuscript)
> - Wording in abstract/introduction is revised to focus more on benchmarking and highlighting existing prior works.
> - Related works section is revised to better include other benchmarking works (such as Shysheya et al., as you mention), around line 485.
>
> We hope that this narrates the paper closer to what is being done.
>
> > Question 1
>
> Thank you for the comment and the detailed derivation. We apologize, as this was a writing error on our part, and one that we noticed after submission. We used the framework and implementation from (Chen et al. 2024), but wanted to write the overview in terms of Gaussian noise $z \sim N(0, I)$ rather than the Wiener process $W_t = \sqrt{t}z$ to be more consistent with the other generative models considered and Albergo et al, 2023. In doing so, we had tried to add an extra factor to the SDE to account for the fact that $\frac{W_t}{\sqrt{t}} = z$, but this had unintentional consequences for the $\gamma$ term.
>
> We have remedied this error and rewritten Section 2 to be in terms of the Wiener process $W_t$ and the coefficient $\sigma(t)$, which is how our code is implemented. In summary, the interpolant becomes:
>
> $$ x_t = (1-t)x_0 + tx_1 +  (1-t)W_t, \quad W_t:=\sqrt{t}z, \quad z\sim \mathcal{N}(0, \mathbf{I}) $$
>
> Where we have made the choices $\alpha(t) = 1-t, \beta(t) = t, \sigma(t) = 1-t$. Also, we use the SDE from Chen et al., 2024, which is:
>
> $$ dx_t^{SDE} = b_\theta(t, x_t, x_0)dt + \sigma(t)dW_t $$
>
> Which uses the correct diffusion coefficient as you mention. We have made this clear next to Equation (4), as suggested in the revised manuscript.
>
> > Question 2, Table 11
>
> Thank you for pointing this out. We indeed use both ODE and SDE-based samplers, and not listing the SDE formulation in Table 11 (now Table 13 in the revised manuscript) was done to make the table concise. We figured that there could be many different samplers for not just SI, but also EDM (ODE/SDE, Euler/Heun methods), so we chose to just write one. However, we recognize that this could be misleading so we make a note that different samplers can be used in the table caption, as well as provide explicit forms for the Euler and EM samplers used below the table, in Equations 13 and 14.
>
> > Question 2, ODE sampling
>
> This is a great observation, and one that we should make more clear. You are right that for stochastic interpolants, there is no distribution for predicted states when using an ODE sampler. Despite this there are a few interesting remarks:
>
> - If there is a need for an output distribution, the sampler can be changed to an SDE sampler to achieve a similar accuracy, but there is a need for more sampling steps.
> - Many PDE systems are deterministic and given enough information about the current state, there is only one possible solution at a close enough future timestep. There are also a class of steady-state, boundary-value problems that are fully deterministic in this sense (Darcy Flow, Linear Elasticity, etc.). This potentially allows ODE-based sampling to be more suited in these cases.
> - Using probabilistic training (i.e., adding random noise to the interpolant states) still helps the model, although it may be sampled deterministically during inference. We ablate this by removing the noise during training to learn a truly deterministic interpolant (Table 9), but find worse performance. Prior work has suggested that using noise in the interpolant helps to mix the source and target distributions and results in a smoother, easier to learn drift.
>
> So you are right that this is a bit of an oddity, but many PDE tasks are fulfilled by a single prediction, rather than a distribution, which is why many of the standard models are deterministic (FNO, Unet, etc.). We have made this more clear in the Appendix (line 1428).
>
> > small comment
>
> Thank you for this note. We have revised our results to mention the similarities between flow matching and stochastic interpolants (around line 286).

---

### Official Review · Reviewer_bB3Y · 2025-11-07

**Soundness:** 3
**Presentation:** 3
**Contribution:** 3
**Rating:** 6
**Confidence:** 4

**Summary:**

The paper proposes a framework for the physics prediction. The problem is defined as learning p(x_t|x_t+1) in a physical system, where x represents the state. The whole process is vae+diffusion/flow models. The key innovation is the diffusion process. Instead of using guassian noise, the model choose x_t as the start.

**Strengths:**

I think the design is reasonable and promising. The choice of vae, DiT, flow/diffusion, works well on the physical prediction.

**Weaknesses:**

I am confused about ODE setting. Assuming for a single x_t, there are two possible x_{t+1}^a and x_{t+1}^b, where does the uncertainty come from? With the same start, the model can only learn the average ( x_{t+1}^a  +  x_{t+1}^b), and the model becomes more likely a purely prediction model with rmse loss: learn the average of the output. During the sampling, the start is fixed, combined with an ODE process, this is impossible to generate different results. Though the result may be good, it is possible the whole process is not a probabilistic model.

**Questions:**

See weakness. I think this question is important to this submission. And I think the biggest challenge for the single start with ODE process is how to get different samples.

---

> ### Author Response · Authors · 2025-11-18
> **Response to Reviewer bB3Y**
>
> Thank you for your review. We hope that we can address your concerns below:
>
> > I am confused about ODE setting. Assuming for a single x_t, there are two possible x_{t+1}^a and x_{t+1}^b, where does the uncertainty come from?
>
> Thank you for pointing this out. For stochastic interpolants, this is a true observation, where there is no uncertainty for a predicted state when using an ODE sampler. Despite this there are a few interesting remarks:
>
> - If there is a need for probabilistic outputs, the sampler can be changed to an SDE sampler to achieve a similar accuracy, but there is a need for more sampling steps.
>
> - Many PDE systems are deterministic and given enough information about the current state, there is only one possible solution at a close enough future timestep. There are also a class of steady-state, boundary-value problems that are fully deterministic in this sense (Darcy Flow, Linear Elasticity, etc.). This potentially allows ODE-based sampling to be faster and more accurate in these cases.
>
> - Using probabilistic training (i.e., adding random noise to the interpolant states) still helps the model, although it may be sampled deterministically during inference. We ablate this by removing the noise during training to learn a truly deterministic interpolant (Table 9), but find worse performance. Prior work has suggested that using noise in the interpolant helps to mix the source and target distributions and results in a smoother, easier to learn drift.
>
> We have made this distinction more clear in our revised manuscript (line 1428). Thanks!

---

### Official Review · Reviewer_qf2T · 2025-11-09

**Soundness:** 3
**Presentation:** 2
**Contribution:** 3
**Rating:** 4
**Confidence:** 4

**Summary:**

This paper benchmarks generative models for deterministic and probabilistic forecasting of physical systems (fluids and weather datasets), contrasting methods that transform from Gaussian noise with those that transform from current states to future states. The authors focus on stochastic interpolants for the latter case and diffusion models/flow matching for the former. They argue that the proximity of successive physical distributions makes stochastic interpolants better suited for forecasting problems, achieving superior accuracy across various metrics.

**Strengths:**

The paper investigates the choice of source distributions (Gaussian noise versus current states) in generative modeling for physical systems, a natural question with practical relevance across many problems. The authors conducted comprehensive benchmark experiments using various evaluation metrics and made useful observations.

**Weaknesses:**

1. The set-up of stochastic interpolants needs more clarification.
First, on page 2, equation 4, the ODE and SDE formulations appear to be inconsistent. The SDE formulation looks incorrect: the drift b already generates the exact marginal distribution at time t=1, so directly adding a stochastic term seems not right. A drift modification should be made to compensate for this additional noise.

In Section 4, it is unclear from the results how SI-E and SI-EM, which correspond to the ODE and SDE formulations respectively, are implemented. The above concern regarding the SDE formulation raises questions about the reliability of these results.

2. For the stochastic interpolant methods, it appears the authors did not feed the current state vector as an additional input to the drift vector b (as seen in Figure 1).

If so, the framework will only guarantee generation of the marginal distribution of u(t+1), rather than the conditional distribution u(t+1)|u(t). When the time lag is small, there is strong correlation between successive states, so the generative model in this marginal setting may still capture some conditional dependence / coupling, particularly for deterministic dynamical systems, so the authors could obtain reasonable accuracy. But in general, this approach appears less mathematically sound than the conditional formulation and requires further clarification.

**Questions:**

1. The diffusion models and flow matching methods use Gaussian source distributions. To generate the conditional distribution u(t+1)|u(t), the authors need to feed the current state u(t) as an additional input to the drift vector. This appears to be what the authors have done based on Figure 1 (Page 2), but it contradicts what is presented in Table 11 (Page 23), where the setup generates the marginal distribution rather than the conditional distribution. The authors provide some explanation on Page 4 under "architecture," specifically noting that "the current state of the system is concatenated to the noisy estimate of the future state as conditioning." I think making this distinction clearer throughout the paper would be helpful.

2. The authors have considered metrics for both deterministic accuracy (i.e., using these models to predict deterministic outputs, such as pointwise values or spectral values—specifically for the Kolmogorov flow and Rayleigh-Benard convection examples in this paper) and probabilistic accuracy (for the climate forecasting example). The latter is natural since these generative models produce inherently probabilistic outputs. For the former, could the authors clarify how the stochasticity in the generative model's output is handled to obtain deterministic predictions? Is it simply averaged to get a point estimate? This is not clear in the current presentation.

3. For the climate emulation example, on page 7 the authors state: "Therefore, while 10-day forecasting errors converge and are reported after 30 epochs, models are fine-tuned for an additional 20 epochs for bias evaluations." Could the authors elaborate on how the fine-tuning and bias reduction are performed? Given the long rollout period required for climatology, short-term prediction methods may reasonably suffer from instability or inaccuracy. Specialized techniques are often needed to address this, and further clarification from the authors would be helpful.

---

> ### Author Response · Authors · 2025-11-18
> **Reponse to Reviewer qf2T**
>
> Thank you for providing the thoughtful review. We have clarified many items in our revised manuscript, which we have uploaded:
>
> > 1. The set-up of stochastic interpolants needs more clarification.
>
> Thank you for pointing this out. This was a writing error on our part, and one that we noticed after submission. We used the framework and implementation from [1], but wanted to write the overview in terms of Gaussian noise $z \sim N(0, I)$ rather than the Wiener process $W_t = \sqrt{t}z$ to be more consistent with the other generative models considered. In doing so, we had mistakenly added an extra factor in the SDE. We have remedied this error and rewritten Section 2 to be in terms of the Wiener process $W_t$. In summary, the interpolant becomes:
>
> $$ x_t = (1-t)x_0 + tx_1 +  (1-t)W_t, \quad W_t:=\sqrt{t}z, \quad z\sim \mathcal{N}(0, \mathbf{I}) $$
>
> We can verify that the noise term $(1-t)\sqrt{t}z$ vanishes at the boundaries and we recover $x_0$ at $t=0$ and $x_1$ at $t=1$. Also, the SDE becomes:
>
> $$ dx_t^{SDE} = b_\theta(t, x_t, x_0)dt + \sigma(t)dW_t $$
>
> Where the noise term ($\sigma(t)dW_t $) is $(1-t)\sqrt{dt}z$, which vanishes as $t\rightarrow 1$. For example, when using an Euler-Maruyama sampler, $\sigma(t)dW_t = (1-t)\sqrt{\Delta t}z$, which injects the most noise when sampling begins at $t=0$ and linearly reduces noise to zero when sampling ends at $t=1$, which is the desired behavior as you mention.
>
> > it is unclear from the results how SI-E and SI-EM, which correspond to the ODE and SDE formulations respectively, are implemented
>
> Thanks for pointing this out. We have expanded on how the ODE and SDE integrators are implemented in Appendix D.1 (line 1403). The SI-E and SI-EM models are the same model (same architecture, weights, training), but just sampled differently. The sampling starts by taking an initial sample $x_0$  from the current state $u(t)$. A drift $dx_\theta(x_t, c, t)$ is calculated at time $t$ with the current sample $x_t$ and conditional information $c$ through the forward pass. The Euler integrator updates the current state according to:
>
> $$ x_{t+1} = x_t + \Delta t * dx_\theta(x_t, c, t)$$
>
> while the Euler-Maruyama integrator updates the current state with:
> $$x_{t+1} = x_t + \Delta t * dx_\theta(x_t, c, t) + \sigma(t)\sqrt{\Delta t} * z$$
>
> where $\Delta t$ is the step size, $\sigma(t)$ is the noise coefficient, and $z \sim N(0, \mathbf{I})$.
>
> > 2. For the stochastic interpolant methods, it appears the authors did not feed the current state vector as an additional input to the drift vector b
>
> Thank you for raising this concern. We apologize for the lack of clarity, we used a conditional stochastic interpolant, but realize that Figure 1 and Section 2 could be misleading. We have updated Figure 1 and the writing in Section 2 (line 98) to make it clear that we implemented and tested conditional generation for all generative models, including stochastic interpolants. In particular, the drift is given the current state $x_0 = u(t)$ as conditioning, resulting in $b_\theta(t,x_t, x_0)$, which is achieved by concatenating $x_0$ to the intermediate states $x_t$ during training and inference.
>
> > Question 1
>
> Thank you for this comment. We have clarified in our revised manuscript that all generative models (DDPM, FM, etc.) are trained for conditional generation. We have updated Table 11 (now Table 13 in the revised manuscript) to show that all models now use conditional information during training and inference. Additionally, we have made this more clear in the main body in Section 3.2 (line 161).
>
> > Question 2
>
> This is a great question. While the generative model output can change every time it is queried, the convention in the PDE field is just to use one of these outputs as the solution estimate and calculate pointwise errors from that single, generated sample and the ground truth. Taking the average of an ensemble of sampled predictions is shown to increase performance (in terms of pointwise RMSE) in PDE/Climate prediction [2, 3], but we do not do this. We have made this more clear in the work (line 205).
>
> > Question 3
>
> Thank you for raising this question. The motivation behind this is that while the training loss does not decrease much after a certain point, the model’s climatological biases can change appreciably. However, as the model continues to train for even longer, the fluctuations in biases eventually tend to dampen, just many epochs after the training losses converge. As you mention, this could be due to a mismatch between the training objective and long-term evaluation. Although we could apply techniques such as unrolled training to optimize for longer-term stability, we instead opt for a simple strategy of just choosing the epoch with the lowest climatological biases for a given model. We will make this more clear in the revised work (line 370).
>
> References:
> 1. https://arxiv.org/abs/2403.13724
> 2. https://arxiv.org/abs/2503.22600
> 3. https://arxiv.org/abs/2312.15796

---

> > ### Comment · Reviewer_qf2T · 2025-11-19
> >
> > Thank you for the clarification.
> >
> > In the updated equation (4), the ODE looks like using the same drift as the SDE. This seems still an issue of inconsistency.

---

> > > ### Author Response · Authors · 2025-11-19
> > >
> > > Thank you for the comment. You are correct that using the same drift in the ODE and SDE is incorrect, which we did not realize. The implications for this are that all the SI or SI-EM results are generated with an incorrect drift, which in this case is very interesting that the ODE sampler is doing so well. However, we have derived an adjusted drift that uses the drift trained for the SDE to create an equivalent ODE. Please see our proof:
> > >
> > > According to [1] (Chen et al., 2024), the conditional drift $b_\theta(t, x_t, x_0)$ for the stochastic interpolant:
> > >
> > > $$  x_t = \alpha(t)x_0 + \beta(t) x_1 + \sigma(t)W_t $$
> > >
> > > can be learned by minimizing the loss:
> > >
> > > $$L_b (b_\theta) = \frac{1}{N}\sum_{i=1}^N \left( |b_\theta(t_i,x^i_{t_i}, x_0^i)- (\dot \alpha (t)x_0^i + \dot \beta(t) x_1^i + \dot{\sigma}(t)W_t )|^2\right) $$
> > >
> > > Samples from the conditional probability density $p(x_t|x_0)$ can be generated by solving the SDE:
> > >
> > > $$dx_t^{SDE} = b_\theta(t, x_t, x_0)dt + \sigma(t)dW_t$$
> > >
> > > This is identical to the SDE setup considered in Section 2 in our work. We consider a case where we would like to sample an equivalent ODE such that the generated samples follow the same distribution as the underlying interpolant and its associated SDE: $Law(x_t^{ODE}) = Law(x_t^{SDE}) = Law(x_t|x_0)$. It will become important later to have values of $\alpha(t), \beta(t), \sigma(t)$, which we have set to $\alpha(t) = 1-t, \beta(t) =t, \sigma(t) = 1-t$, consistent with Section 2.
> > >
> > > Again, from Chen et. al. [1], the learned drift $b_\theta(t, x_t, x_0)$ can also be manipulated to define a family of SDEs:
> > >
> > > **Chen et. al., 2024, Theorem 3.2:**
> > > > Given any $g \in C^0[0, 1]$ such that $lim_{t\rightarrow0^{+}} \frac{g(t)^2 - \sigma(t)^2}{t}$ and $lim_{t\rightarrow1^{-}} \frac{g(t)^2}{\sigma(t)}$ exists, define: $$\hat b_\theta(t, x_t, x_0) = b_\theta(t, x_t, x_0)  + \frac{1}{2}(g_t^2 - \sigma_t^2) \nabla \log p(x_t|x_0)$$
> > > where $b_\theta(t, x_t, x_0)$ is the minimizer of the loss defined above. Then the solutions to the SDE:
> > > $$d\hat x_t^{SDE} = \hat b_\theta(t, x_t, x_0)dt + g(t)dW_t $$
> > > are such that $Law(\hat x_t^{SDE}) = Law(x_t^{SDE}) = Law(x_t|x_0)$ for all $t, x_0$.
> > >
> > > Interestingly, we can set $g(t) = 0$ to change the above SDE into an ODE:
> > > $$ d\hat x_t^{ODE} = \hat b_\theta(t, x_t, x_0)dt $$
> > >
> > > One of the regularity conditions is not met by this choice of $g(t)$ ($lim_{t\rightarrow0^{+}} \frac{g(t)^2 - \sigma(t)^2}{t}$), however, we will see later how this can be remedied by our particular choice of $\alpha(t), \beta(t), \sigma(t)$. For now, what remains is to derive the modified drift $\hat b_\theta(t, x_t, x_0)$ by calculating the score function $\nabla \log p(x_t|x_0)$. From Chen et. al. [1], the score function can be expressed as:
> > >
> > > $$ \nabla \log p(x_t|x_0) = A_t(\beta(t)b_\theta(t, x_t, x_0) - c_t(x_t, x_0)) $$
> > > where
> > > $$ A_t= \frac{1}{t\sigma(t)(\dot \beta(t)\sigma(t) - \beta(t)\dot\sigma(t))}, \quad c_t(x_t, x_0) = \dot\beta(t)x_t + (\beta(t)\dot\alpha(t) - \dot \beta(t)\alpha(t))x_0 $$
> > > Substituting in our choices for $\alpha(t) = 1-t, \beta(t) =t, \sigma(t) = 1-t$ results in:
> > > $$ A_t = \frac{1}{t(1-t)}, \quad c_t(x_t, x_0) = x_t - x_0 $$
> > >
> > > Therefore the score function becomes:
> > > $$ \nabla \log p(x_t|x_0) = \frac{1}{t(1-t)}(tb_\theta(t, x_t, x_0) - (x_t - x_0)) $$
> > >
> > > and the new drift for the ODE becomes:
> > > $$ \hat b_\theta(t, x_t, x_0) = b_\theta(t, x_t, x_0)  + \frac{-(1-t)}{2t}(tb_\theta(t, x_t, x_0) - (x_t - x_0)) $$
> > >
> > > The regularity condition $lim_{t\rightarrow0^{+}} \frac{g(t)^2 - \sigma(t)^2}{t}$ exists in order to make the modified drift $\hat b_\theta(t, x_t, x_0)$ exist at $t=0$ for all choices of $\alpha(t), \beta(t), \sigma(t)$ (Chen et al., 2024, Appendix B.4). However given our coefficients, we can see that at $t=0$, $x_t-x_0 = 0$ and therefore:
> > >
> > > $$ \hat b_\theta(0, x_t, x_0) = b_\theta(0, x_t, x_0)  + \frac{-1}{2}b_\theta(0, x_t, x_0)$$
> > >
> > > This expression can be analytically solved during the first integration step at $t=0$ to avoid singularities in the ODE.
> > >
> > > Therefore, without retraining $b_\theta(t, x_t, x_0)$, we have access to an ODE that samples the same distribution defined by the SDE, and perhaps gives some insight into why the original formulation works. However, we are now rerunning all our SI and SI-E results with this new drift and an Euler sampler, and should hopefully have a revised manuscript ready by this weekend. Preliminary results suggest that the new drift is actually more accurate than the old drift when using an ODE sampler (likely since the drift is actually correct now), however, more sampling steps are needed likely since the new ODE does not define straight paths with respect to $b_\theta(t, x_t, x_0)$ any more. Thank you for your patience and also for alerting us to this issue! Please let us know if you have any concerns with this as well.

---

> > > > ### Author Response · Authors · 2025-11-24
> > > >
> > > > Dear reviewer,
> > > >
> > > > Thank you for your patience during our revision. We have updated our manuscript with revised results using the new ODE formulation. The main points are:
> > > > - Tables and figures with SI-E have been updated
> > > > - A new Appendix A has been created for the above derivation
> > > > - Section 2 is modified to discuss this ODE formulation
> > > >
> > > > We will be transparent by pointing out some limitations of this ODE formulation. Solving the ODE allows us to numerically generate a sample from $p(x_1|x_0)$, however it does not model a distribution. The diffusion term in the SDE allows transport from a Dirac delta $p(x_0)$ to a distribution $p(x_1|x_0)$, while the ODE represents the limiting case when the diffusion term approaches zero. However, in practice, this method still produces predictions with low errors. We hope that as a benchmark study, the value lies in comparisons to other generative models/datasets and the empirical takeaways, and future work can perhaps investigate this further.

---

### Author Response · Authors · 2025-12-01
**Overview of Review/Revisions**

Dear reviewers and AC,

We just wanted to provide a quick summary of the comments and revisions to aid in the decision process. We will try to be as concise and neutral as possible:

- Concerns about correctness/notation: We originally made an error in the writing which resulted in some inconsistencies. These have since been remedied.
- Concerns about scope/message: We agree with these concerns and have altered the paper to focus on benchmarking stochastic interpolants. The title/framing of the paper has been changed.
- Errors with the ODE formulation: This was an error pointed out to us that we were not originally aware of. We have derived a fix and rerun all experiments that used an ODE sampler. The results are mostly the same.
- Concerns about the paper's motivating evidence: These are valid, and we took the time to more properly explain our motivations as well as scale back the experiment to a simpler system. It seems that the reviewer is satisfied with this.
- Concerns about rigor of baselines: We have performed more hyperparameter tuning for the baselines and tried to be as fair as possible in our experiments. It seems that the reviewer is satisfied with this as well.

We want to thank the AC and reviewers for their time. Regardless of the outcome, we hope that everyone will have a nice holiday season.

---

### Meta-Review · Area_Chair_33Rf · 2025-12-20

**Summary:**

This paper presents a comprehensive benchmark of stochastic interpolants (SI) as surrogates for physical systems, specifically focusing on autoregressive forecasting in PDEs and climate modeling. The core hypothesis is that mapping directly between successive physical distributions is more efficient and accurate than traditional methods that transport Gaussian noise to a target state.

The meta-review decision is informed by the following key concerns raised during the process:

- Technical Correctness: Initial versions of the paper contained significant errors in the SDE/ODE formulations and notation.
- Novelty vs. Benchmarking: Reviewers questioned whether the work provided a "reframing" or new methodology, noting that the application of SI to fluids is not entirely new.
- Motivational Evidence: There were contradictions in the initial distance heuristic results (SWD/MMD vs. C2ST) and concerns about the stationarity of the physical systems used to justify the approach.
- Fairness of Comparison: One reviewer argued that SI's superior performance might be due to unequal hyperparameter tuning compared to the baselines.

My own concerns are:

-  While the paper evaluates stochastic interpolants (SI) on autoregressive tasks, the "benchmark" focuses heavily on relatively short-term lead times (e.g., 10 days for climate or $t=50$ for fluids).
- One of the primary selling points of SI is efficiency (fewer sampling steps). If the mathematically rigorous ODE formulation requires high Number of Function Evaluations to be accurate, the efficiency advantage over Flow Matching or EDM-E may diminish or vanish in practice.
- While the abstract and introduction suggest that SI can learn maps between arbitrary densities, no corresponding experiments are observed in the manuscript.

**Reviewer Concerns:**

Addressed concerns:

- Mathematical Errors: The authors successfully derived a corrected "adjusted drift" for the ODE formulation and updated their experimental results. This fixed the inconsistency between SDE and ODE paths.
- Conditioning: The authors clarified that all models, including SI, are trained for conditional generation by concatenating the current state to the input.
- Positioning: The authors effectively pivoted from "reframing the field" to a "benchmarking study," changing the title and abstract accordingly.
- Distance Heuristics: The authors replaced the problematic Rayleigh-Bénard distance plots with a more rigorous analysis of Kolmogorov Flow, which has a better-understood stationary distribution.

Outstanding concerns:

- Statistical Robustness: While the authors explained their computational constraints (6 A100-days per run), the complex climate and RBC results still lack error bars/multiple seeds, which remains a statistical limitation.
- Innovation: The work remains primarily an empirical study. While valuable, there is no fundamental methodological "breakthrough" proposed.

**Reviewer Scores:**

- Todc (2): They explicitly stated the rebuttal addressed some concerns regarding validity and fairness, but novelty and error bar issue remain.
-  qf2T (4 to 6): Although they started at 4 (marginal), the authors provided a rigorous proof for the adjusted drift they requested.
- bB3Y (6): Their confusion over the ODE/uncertainty was addressed by the authors' explanation of deterministic PDE systems and the role of noise in training vs. sampling.
- KFnG (6): Their detailed technical feedback was integrated into the revised Appendix A and Section 2.

---

### Decision · Program_Chairs · 2026-01-26

Reject